# Research

theoretical biology, ecology, evolution

invasion, dispersal, evolution, trade-off, population dynamics, mathematical model

**Author for correspondence:**
Stephen J. Cornell
e-mail: stephen.cornell@liverpool.ac.uk

†Present address: Department of Mathematics and Statistics, University of Strathclyde, Livingstone tower, 26 Richmond Street, Glasgow G1 1XH, UK.

# Anomalous invasion dynamics due to dispersal polymorphism and dispersal–reproduction trade-offs

Vincent A. Keenan† and Stephen J. Cornell

Institute of Integrative Biology, University of Liverpool, Crown Street, Liverpool L69 7ZB, UK

VAK, 0000-0002-2307-0680; SJC, 0000-0001-6026-5236

Dispersal polymorphism and mutation play significant roles during biological invasions, potentially leading to evolution and complex behaviour such as accelerating or decelerating invasion fronts. However, life-history theory predicts that reproductive fitness—another key determinant of invasion dynamics—may be lower for more dispersive strains. Here, we use a mathematical model to show that unexpected invasion dynamics emerge from the combination of heritable dispersal polymorphism, dispersal-fitness trade-offs, and mutation between strains. We show that the invasion dynamics are determined by the trade-off relationship between dispersal and population growth rates of the constituent strains. We find that invasion dynamics can be 'anomalous' (i.e. faster than any of the strains in isolation), but that the ultimate invasion speed is determined by the traits of, at most, two strains. The model is simple but generic, so we expect the predictions to apply to a wide range of ecological, evolutionary, or epidemiological invasions.

## 1. Introduction

Evolution can significantly affect the dynamics of biological invasions and range expansion [1–3]. Deleterious mutations can accumulate during invasions, leading to a deceleration in the rate of advance [4], but evolutionary processes can also facilitate invasions. Dispersal evolution has attracted particular interest, as higher dispersal ability confers a clear advantage when colonizing new habitat [3]. Empirical data suggest that newly established populations contain individuals with elevated dispersal capabilities [5,6], and accelerating cane toad invasions are accompanied by an evolution of dispersal-related traits such as leg length [7] and dispersive behaviour [8]. The tendency of more dispersive strains to be found at the vanguard of an invasion has been dubbed 'spatial sorting' [9], a form of dispersal evolution that plays out in space as well as time [10].

However, invasion dynamics depend on population growth as well as dispersal [11,12]. A higher investment in dispersal ability can imply a lower investment in traits related to reproduction [13], so an increase in dispersal ability does not necessarily imply a faster invasion. Moreover, a sufficiently strong trade-off could disrupt strict spatial sorting—a hypothetical example would be a highly dispersive but infertile strain, which would not advance the invasion. Simulation studies show that the rate of spread can accelerate in the presence of a dispersal-fecundity trade-off [14], but to predict the general conditions under which different invasion scenarios (e.g. spatial sorting, constant or accelerating speed, etc.) occur requires a mathematical theory. Several studies have shown that invasion speed for a polymorphic species with a trade-off is determined by the strain which would invade most quickly on its own [10,15,16], but those studies omit mutation—which plays a key role in invasion dynamics by creating interactions between strains. Elliott & Cornell [17] showed that, if a species comprises two strains (one a superior disperser, and one more fecund), with infrequent mutation

between strains, it can invade at a faster 'anomalous' speed than would be predicted for either strain in isolation. However, it is not clear what would occur in a more realistic species consisting of multiple strains. For example, it is not obvious whether the invasion dynamics are determined by all of the strains or just a few, or how the invasion dynamics could be predicted for a species with a very large number of strains. Since both dispersal and reproductive fitness are beneficial, invasions could potentially promote a mixed strategy or even evolutionary branching [18,19]. Here, we define 'reproductive fitness' as per capita growth rates at low density and use this definition throughout the article.

We study a model for invasions by species with dispersal polymorphism, a dispersal-fitness trade-off, and mutation between strains. We show that the shape of the trade-off curve critically determines whether the invasion speed continues to accelerate or approaches an asymptote. The trade-off curve also determines whether the invasion speed equals that of one of the constituent strains, or whether the speed is anomalous, i.e. is faster than any single strain on its own. Surprisingly, in all cases, we find that the asymptotic speed is determined by the traits of, at most, two of the constituent strains. We find that strong effects of spatial sorting are not realized in all cases—the most dispersive strains do not necessarily lead the invasion—and that the invasion speed is not necessarily determined by the most dispersive or the most fecund strain, or the one which, in isolation, would invade at the fastest speed. Our model is simple but generic and requires large population sizes, so is most likely to apply to microbial or viral systems such as infectious diseases.

## 2. Methods

We consider an asexual haploid species consisting of $N$ strains (i.e. genotypes), where each strain has a distinct growth rate and dispersal phenotype, with the possibility of mutation between strains at birth (derivation can be found in electronic supplementary material, appendix S1). We first consider a deterministic model in continuous space and time, for which we can derive exact results. We assume density-dependent competition between strains and that dispersal can be approximated by diffusion, and model the dynamics using the following spatial Lotka–Volterra model

$$\frac{\partial n_i}{\partial t} = D_i \frac{\partial^2 n_i}{\partial x^2} + r_i n_i \left(1 - \sum_{j \in S} C_{ji} n_j\right) + \sum_{j \in S \setminus \{i\}} \mu(v_{ji} n_j - v_{ij} n_i), \tag{2.1}$$

for all $i \in S$, where: $S = \{1, 2, \ldots, N\}$ is the set of all strains; $n_i$, $D_i$ and $r_i$ are, respectively, the density, diffusion constant, and population growth rate of the $i$th strain, $C_{ji}$ is the competitive effect of strain $j$ on $i$, and $\mu v_{ij}$ the mutation rate from strain $i$ to strain $j$. This extends the investigation of Elliott & Cornell from two strains to a general degree of population polymorphism [17]. Equations including diffusion, Lotka–Volterra competition, and mutation between strains have been considered before [20–22], but explicit results for the case where both $D_i$ and $r_i$ differ among strains have not previously been computed for $N > 2$. We assume that the species does not comprise sub-species between which mutation is not possible (i.e. that the $v_{ij}$ allow an individual to have descendents of any strain, after enough generations), and that the $C_{ij}$ are such that there is a single stable spatially uniform equilibrium.

We consider an invasion, where half of space is initially empty and the other half is occupied by the species at equilibrium, in

which case solutions to equations (2.1) converge to travelling waves of the form $n_i = N_i(x - c^*t)$. Girardin [22] has proven that the spreading speed $c^*$ for equations of the form (2.1) is given by linearizing around the unstable equilibrium $n_i = 0$ and finding the solution of the form $n_i \sim \exp(\lambda t - kx)$ whose phase velocity $\lambda/k$ is smallest. We compute an exact expression for $c^*$ in the biologically interesting case of weak mutation ($\mu \to 0^+$). More details of the calculation, and a graphical illustration, can be found in electronic supplementary material, appendix S1.

It is known that demographic stochasticity can affect anomalous speeds in dimorphic species [23], so to test the robustness of our results we also ran simulations of a stochastic Beverton–Holt model. This is based on the following deterministic model:

$$m_i(x, t) = n_i(x, t)(1 + r_i(1 - N\eta)) + \eta \sum_{j \in S} r_j n_j(x, t) \tag{2.2}$$

$$q_i = \frac{m_i}{1 + a \sum_j m_i} \tag{2.3}$$

$$n_i(x, t+1) = q_i(x, t)(1 - 2D_i) + D_i(q_i(x - 1, t) + q_i(x + 1, t)), \tag{2.4}$$

where equation (2.2) represents reproduction with mutation, equation (2.3) density-dependent mortality, and equation (2.4) dispersal. However, in the stochastic model Poisson and Multinomial pseudorandom number generators replace equations (2.3) and (2.4) (more details in the electronic supplementary material, appendix S1). Here, $r_i$ and $D_i$ again represent population growth rate and dispersal ability, and $\eta (< 1/N)$ is a mutation parameter. The parameter $a$ sets the scale of density dependence, so that the stable equilibrium density is proportional to $1/a$. We expect that the stochastic and deterministic version of the model will behave similarly when $a$ is very small.

## 3. Results

We find that anomalous invasion speeds are possible in the $N$-strain case, just as was found in the 2-strain case (figure 1). When the mutation rate is very small but non-zero (i.e. in the limit $\mu \to 0^+$), the speed of invasion by any $N$-strain species is obtained by calculating the maximum of $c_i^m$ and $c_{ij}^d$ in the following expressions:

$$c_i^m = 2\sqrt{r_i D_i}, \tag{3.1}$$

for all $i$, and

$$c_{ij}^d = \frac{|r_i D_j - r_j D_i|}{\sqrt{(r_i - r_j)(D_j - D_i)}}, \tag{3.2}$$

for all pairs $i$ and $j$ such that

$$\frac{r_i}{r_j} + \frac{D_i}{D_j} \geq 2 \quad \text{and} \quad \frac{r_j}{r_i} + \frac{D_j}{D_i} \geq 2. \tag{3.3}$$

Here, $c_i^m$ is the 'monomorphic speed' at which a species consisting solely of strain $i$ would invade. $c_{ij}^d$ is the 'dimorphic speed' at which a species consisting solely of strains $i$ and $j$ would invade, provided conditions (3.3) are met [17]. Conditions (3.3) imply that valid dimorphic speeds only exist for pairs of strains whose dispersal and growth rates differ sufficiently—a graphical representation can be seen in [17] figure 1. If the largest valid $c_{ij}^d$ is greater than the largest $c_i^m$, then the invasion is more rapid than that for any of the constituent strains in isolation, and the invasion is said to be 'anomalous' [17].

Our analysis shows that an $N$-strain species therefore invades at the same speed as if it consisted of only one or

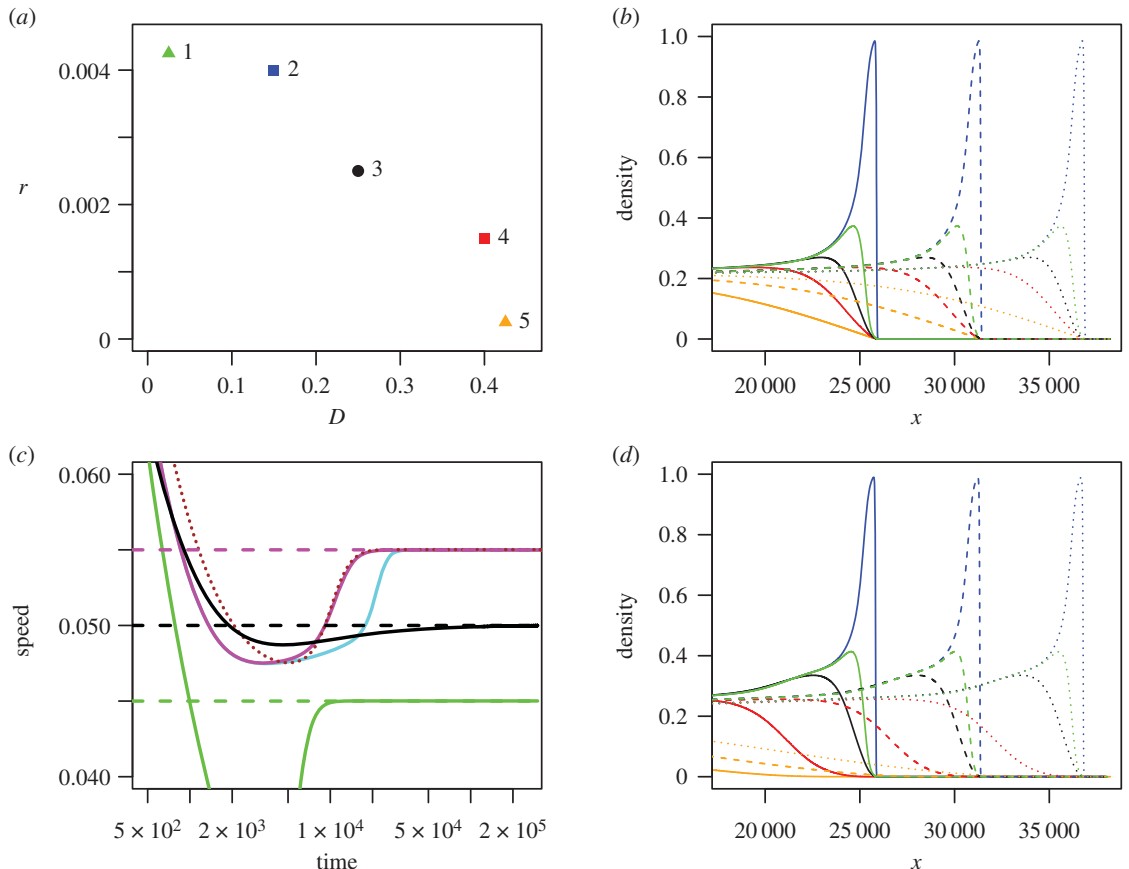

**Figure 1.** Anomalous invasions by polymorphic species into unoccupied habitat, where all strains can coexist at equilibrium, from numerical solutions of equation (2.1). Panel (*a*): trait values for five strains used in this figure. $(D_i, r_i) = (0.025, 0.00425), (0.15, 0.004), (0.25, 0.0025), (0.4, 0.0015), (0.425, 0.00025)$ for strains $i = 1, 2, 3, 4,$ and 5, respectively. Squares denote the 'vanguard' strains (2 and 4) and triangles the 'extreme' strains with the highest growth rate (strain 1) and highest dispersal ability (strain 5), respectively. Strain 3 (circle) has the highest monomorphic speed. Colours correspond to those used in (*b,d*) (green, blue, black, red, and orange for $i = 1, 2, 3, 4,$ and 5, respectively). (*b,d*) Population density as a function of spatial coordinate $x$ during invasion by a species consisting of all five strains, displaying travelling waves, at times $t = 4 \times 10^5$ (solid curves), $5 \times 10^5$ (dashed curves), $6 \times 10^5$ (dotted curves). Mutation is universal among strains ($\nu_{ij} = 1 \forall i, j$) in panel *b*, but in panel *d* is only non-zero between neighbouring strains ($\nu_{ij} = 1$ if $|i - j| = 1$, 0 otherwise). (*c*) Speed of position of front for species consisting of different combinations of strains. Magenta: all five strains present, universal mutation. Cyan: all five strains present, neighbouring-strain mutations only. Brown dotted curve: vanguard strains (2 and 4) only. Black: strain 3 (fastest monomorphic speed) only. Green: extreme strains (1 and 5) only. Dashed horizontal lines are theoretical predictions for $\mu \to 0$: $c = c_{24}^d = 0.055$ (magenta); $c = c_3^m = 0.05$ (black); $c = c_{15}^d = 0.045$ (green). In all cases, mutation rate $\mu = 10^{-6}$, and competition coefficients are $C_{ij} = 1$ for $i = j$, 0.9 otherwise.

two of its constituent strains—the traits of the other strains do not affect the invasion speed. This is illustrated by simulation in figure 1 for a species consisting of five strains. Parameters are chosen (figure 1*a*) so that the strains that are predicted to determine the invasion speed (2 and 4) are not the strain with the highest monomorphic speed (strain 3) nor the strains with the highest population growth rates or dispersal (strains 1 and 5). Figure 1*c* shows that, after a transient, the invasion for a 5-strain species (magenta curve) advances at the same speed as a species containing only strains 2 and 4 (brown dotted curve), which are the strains with the highest dimorphic speed $c_{ij}^d$ from equation (3.2). This is faster than either strains 2 or 4 in isolation (not shown), or the strain with the fastest monomorphic speed in isolation (black curve). Thus, while anomalous invasion dynamics implies a synergy between two strains (benefiting from the growth rate of a more fecund strain and the dispersal ability of a more dispersive one), this synergy does not extend beyond more than two strains. Moreover, one might expect that the 'vanguard' strains (i.e. the two strains whose traits determine the anomalous speed) would be the most dispersive and the most fecund, so that the population as a whole benefits from the highest diffusion constant and the

highest population growth rate. However, it turns out that this need not be the case: a species consisting of strains 1 and 5, which have, respectively, the highest fecundity and dispersal ability, invades more slowly (green curve in figure 1*c*). Note from figure 1*b* that the invasion shows weak effects of spatial sorting: the invasion is led by the second least dispersive strain (blue curve), and the strains with the highest monomorphic speed (black) and dispersal (orange curve) trail behind. In all cases, the long-term invasion speeds closely match the predictions for small mutation rate from equations (3.1) and (3.2) (horizontal dashed lines in figure 1*d*).

Girardin's proof [22] shows that the invasion speed is determined by the dynamics at low densities, and is therefore independent of the competition coefficients $C_{ji}$. Thus, we obtain the same anomalous speeds whether or not the strains coexist within the range core (i.e. at the stable equilibrium). In figure 2*a*, the invasion speed of a species consisting of strains 2, 3, and 4 (magenta) is given by the anomalous speed for strains 2 and 4 (brown dotted curve), even though both are outcompeted by strain 3 in the core range. This shows that strains that are very rare in the range core of the species can still determine the invasion dynamics. Strain 2, which

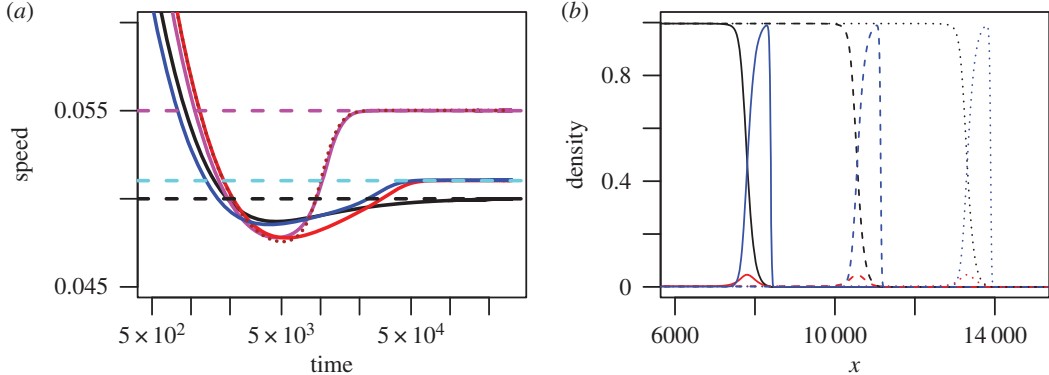

**Figure 2.** Strains that are outcompeted at equilibrium can still generate anomalous invasion speeds. Strains are numbered as in figure 1a, but competition coefficients are such that strain 3 outcompetes strains 2 and 4 at equilibrium ($C_{ji} = 0.8$ for $i = 3$ and $j = 2$ or 4; $C_{ji} = 1.25$ for $j = 3$ and $i = 2$ or 4). (a) Invasion speeds as a function of time. Colours denote different combinations of strains being present in the species: strains 2,3,4 with universal mutation (magenta); strains 2 and 4 (brown dotted); strains 2 and 3 (blue); strains 3 and 4 (red); strain 3 only (black). Dashed horizontal lines are theoretical predictions for $\mu \to 0$: $c = c_{24}^d = 0.055$ (magenta); $c = c_{23}^d = c_{34}^d = 0.05103$ (cyan); $c = c_3^m = 0, 5$ (black). (b) Population density as a function of spatial coordinate $x$, at times $t = 10^5$ (solid curves), $1.5 \times 10^5$ (dashed curves), $2 \times 10^5$ (dotted curves). In b, strains are coloured as in figure 1a. From numerical solutions of equation (2.1), with mutation $\mu = 10^{-6}$.

has the highest population growth rate, leads the front with high density (figure 2b, blue curve). Strain 4, which has the highest dispersal ability, has a higher density in the wake of the front than at equilibrium but lower density than the other two strains (figure 2b, red curve). Nevertheless, removing either strain 2 or 4 slows the invasion (figure 2a, red and blue curves), to a slower but still anomalous speed (cyan horizontal dashed line), which is quicker than the invasion speed for strain 3 alone (black), which in turn is quicker than either vanguard strains 2 or 4 in isolation (not shown).

It is important to note that anomalous speeds require mutation to be non-zero, but persist even when the mutation rates between strains are vanishingly small (i.e. in the limit $\mu \to 0$, but still non-zero). This is surprising because, when the mutation rates are strictly zero, anomalous speeds do not occur and the strains invade independently at their monomorphic speeds (or not at all, if they are outcompeted in the stable equilibrium [16,17]). In the partial differential equation model (2.1), a small amount of mutation, combined with exponential growth, is sufficient for the different strains to keep up with each other during the invasion and participate in the invasion. However, we will show later that demographic stochasticity eliminates anomalous invasion speeds unless $\mu$ is sufficiently large (figure 4).

Furthermore, the limiting invasion speed when $\mu \to 0$ does not depend on the relative values $v_{ij}$ of the mutation rates, even if some are zero (provided the system does not factorize into independent subsets of strains between which mutation is impossible). In particular, the invasion speed is the same for a system with 'universal' mutation ($v_{ij} = 1$ for all $i$ and $j$) as for 'nearest-neighbour' mutation ($v_{ij} = 0$ unless $|i - j| = 1$) (figure 1d, cyan curve in figure 1c). This means that our results should not only apply to species with a small set of discrete strains, but also extend to the case of a very large set of strains where traits mutate by small amounts at each generation.

This allows us to predict the invasion speed for a species with a continuous set of strains. We assume that each strain has a unique phenotype determined by $r$ and $D$, and that there is a trade-off between $r$ and $D$ so that $r(D)$ is a decreasing function of $D$. The invasion speed, after a sufficiently long time, will again be given by the largest permitted value of $c_i^m$,

and $c_{ij}^d$ calculated using equations (3.1)–(3.3) for all strains, and pairs of strains of the species. These equations have a geometric interpretation and the invasion speed depends, in a simple way, on the shape of the $r(D)$ curve. In particular, the existence of an anomalous invasion speed is determined by the curvature of the trade-off curve (figure 3; electronic supplementary material, appendix S1).

Some simple algebra (electronic supplementary material, appendix S1) shows that the dimorphic speed for two strains is equal to the monomorphic speed for a 'virtual' strain, which lies at the midpoint (green triangles in figure 3) of the straight-line segment joining the two axes (dotted line in figure 3) and passing through the points representing the two strains (blue circles in figure 3) in $r(D)$–$D$ space. Further simple algebra shows that, if the virtual strain lies between the two real strains, then conditions (3.3) are met and this is a valid dimorphic speed (figure 3a,c,e,f); otherwise, this is not a valid dimorphic speed for the species.

Therefore, if the trade-off curve is a straight line (figure 3a,b), then all pairs of strains have the same virtual strain (and therefore the same dimorphic speed) whether or not the trade-off curve encompasses this virtual strain. In both figure 3a,b, the invasion speed will equal the monomorphic speed of the fastest constituent strain. On the other hand, if the trade-off curve has negative curvature, figure 3e, then the virtual strain for any pair of strains either lies below the trade-off curve, or does not lie between the two real strains. In that case, none of the valid anomalous speeds are faster than the fastest monomorphic speed for the species. In both of these cases, the ultimate invasion speed will be the same as the monomorphic speed for the fastest strain in isolation, i.e. the strain with the highest value of $Dr(D)$ (see equation (3.1)).

However, if the trade-off curve has positive curvature (figure 3c,d), then the chord joining the two most extreme strains will lie above the trade-off curve. If the range of $r$ and $D$ values is wide enough that the corresponding virtual strain lies between these extreme strains (figure 3c), then this will have the fastest dimorphic speed, which will be faster than any of the constituent monomorphic speeds. However, if the range of $r$ and $D$ values are not wide enough (figure 3d), the species will invade at the fastest monomorphic speed. In the former case, the invasion will follow the anomalous speed when the

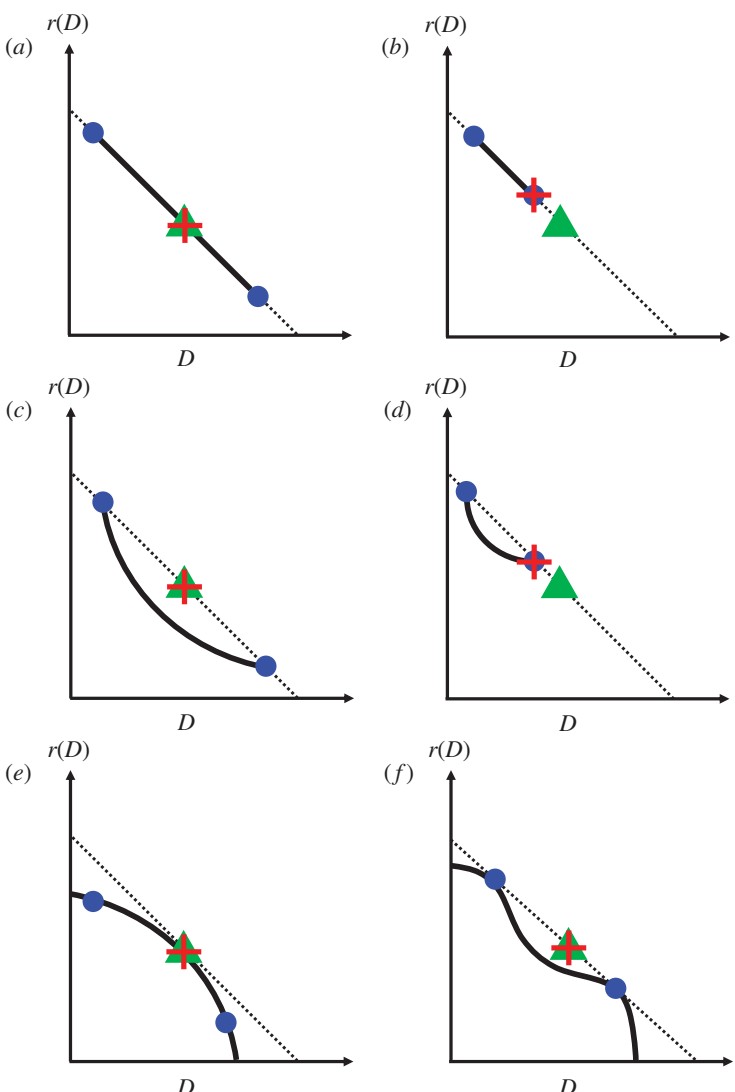

**Figure 3.** The geometry of the $D$–$r$ trade-off determines which anomalous speeds, if any, give the eventual invasion speed for a species comprising a continuum of strains. Solid lines represent the trade-off curve between $D$ and $r$. Dotted lines are chords that pass through points on the trade-off curve and terminate at the axes. Green triangles are the midpoint of these chords, and represent the 'virtual strain' for any two strains that the chord passes through (see text). Blue points represent two particular strains of the species. The invasion speed in each case is given by $2\sqrt{D_+ r_+}$, where $D_+$ and $r_+$ are the values of $D$ and $r$ evaluated at the red crosses. Panels $a,b$: for a straight-line trade-off, all pairs of species have the same chord and therefore the same virtual strain. In $a$, the virtual strain is a constituent of the species, but not in $b$. In $c$, the virtual strain for the two extreme strains (blue circles) lies between them, and since the trade-off curve lies below this chord the virtual strain has a faster speed than any single constituent. In $b,d$, the virtual strain does not lie between any pair of points on the trade-off curve, so does not yield a valid anomalous invasion speed. If the trade-off has negative curvature ($e$) there are valid anomalous speeds, but none of their virtual strains lie above the trade-off so the asymptotic invasion speed is the fastest constituent monomorphic speed. In $f$, the chord is tangential to the trade-off at the two blue points, and since the whole trade-off curve lies below this chord the corresponding anomalous speed is faster than any other monomorphic or dimorphic speed for this species. This shows that the vanguard strains are not necessarily the ones with the highest values of $r$ or $D$. We show further in the Methods section that the vanguard strains do not necessarily have the highest value of $rD$ either.

vanguard strains have the highest $r$ and the highest $D$. A further possibility is that the curvature of the trade-off curve is positive in some places and negative in others (figure 3$f$), for example, if the trade-off is more acute at more extreme values. In this case, the vanguard strains will be the ones where the chord joining them is tangential to the trade-off at both points, which will not represent the most extreme traits in the species (similar to what was found in figure 1 for a species with a discrete set of trait values, where the vanguard strains can be found from figure 1$a$ using the same construction as figure 3$f$).

While the deterministic model, equation (2.1), predicts anomalous speeds for vanishingly small mutation, it has been shown for $N = 2$ that demographic stochasticity suppresses anomalous speeds when mutation or populations are small [23]. We also find this to be the case for $N > 2$. While

figure 4$b$ shows that a 5-strain species (black data) always invades at a speed close to that for a 2-strain species consisting of the vanguard strains (magenta data), this is only faster than the fastest monomorphic speed (blue data) if either the mutation rate is high or local populations (which are proportional to $1/a$) are large. We again find that the vanguard strains (labelled 7 and 9 in figure 4$a$) need not be the most dispersive or the most fecund if the curvature of the dispersal-fitness trade-off resembles figure 3$f$.

## 4. Discussion

We have extended existing theory for invasions by a dimorphic species where dispersal ability varies among lineages and

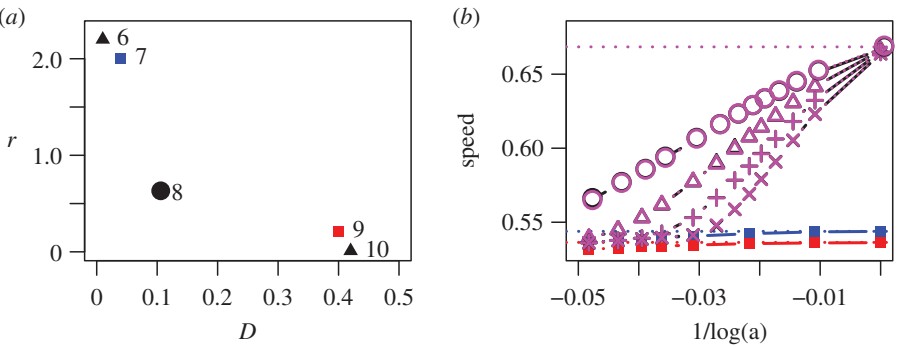

**Figure 4.** Effect of demographic stochasticity on anomalous invasion speeds. Panel *a*: trait values for five strains, where $(D_i, r_i)$ = (0.01, 2.2), (0.04, 2), (0.1, 0.6), (0.4, 0.21), and (0.42, 0.01) for $i$ = 6, 7, 8, 9, and 10, respectively. Strains 7 (the establisher, blue) and 9 (the disperser, red) are the vanguard strains. Panel *b*: invasion speeds as a function of the strength $a$ of density dependence (which is inversely proportional to the carrying capacity) for species consisting of different combinations of the strains in panel *a*. Red: strain 9 only; blue: strain 7 only; magenta dotted: strains 7 and 9 only; black (partly obscured by the magenta data): all five strains together, with universal mutation. For the polymorphic species, different symbols denote the mutation parameter $\eta$: $10^{-3}$(circles), $10^{-4}$(triangles), $10^{-5}$(plus signs), $10^{-6}$ (multiplication signs). Horizontal dotted lines are predicted speeds (see electronic supplementary material, appendix S1) in the deterministic limit $a \to 0$ for the polymorphic species (magenta), strain 7 alone (blue), and strain 9 alone (red). From simulations of the stochastic discrete model.

trades off against fitness to the case of $N$ strains. In some cases, invasions advance at the same rate as if the species consisted solely of the single strain that has the highest monomorphic speed. However, if the dispersal-growth curve has positive curvature, and extends over a wide enough range of dispersal rates, then an anomalous invasion occurs which is faster than that for any single constituent strain. This speed depends only on the traits of two 'vanguard' strains, which need not be the most dispersive nor the most fecund. The invasion speed is insensitive to the mutation rates between strains (provided they are small but non-zero) or the details of inter-strain competition (as proven by Girardin [22]), and anomalous speeds can even be caused by strains that are outcompeted at equilibrium (figure 2).

Our results are counterintuitive in a number of respects. First, while synergy between strains can cause polymorphic species to behave differently from monomorphic species [24], it is surprising that the effect is still strong (figures 1*c* and 2*b*) even when mutation—the process generating this synergy—is very weak. In other examples of anomalous invasion speeds, each strain has a strongly positive effect on the other [25,26], and it has not been shown whether the anomalous speedup would remain strong if the cooperative interactions were weak. Note that [26] use 'anomalous' in a more restrictive sense than we do, requiring that at least one component of the system spreads at a slower speed; our model does not display 'anomalous' speeds *sensu* Weinberger *et al.* [26] because all strains spread at the same speed, but their models do display anomalous speeds according to our definition. Note also that there are other examples where a vanishingly small parameter can have a non-vanishing effect on an invasion speed, such as when a small fraction of a population disperses according to a fat-tailed dispersal kernel (see Lutscher [27], ch. 12). Second, since the species exploits the dispersal ability of one strain and the fecundity of another, it is surprising that the vanguard strains do not necessarily have the highest dispersal or reproductive ability. Third, if strains interact synergistically, then one might expect that more than two strains would contribute. A similar analysis can be applied to any model where invasion dynamics are determined by the linearized behaviour at low density (i.e. 'pulled waves', [28–30]), and the ultimate invasion speed for low mutation rate will be given either by the

minimum of the $\lambda/k$ curve for one strain, or the intersection between the curves of two strains (see electronic supplementary material, appendix S1 figure 1). Thus, no matter how many traits trade off against each other, or whatever density dependence acts at higher density, we predict that anomalous invasion speeds at low mutation rate depend only on the traits of two strains.

Our results show that dispersal evolution during invasion is still more complex than previously thought. Evolving invasion fronts are not necessarily led by the most dispersive strains [5,9,31,32], nor even by the one that would invade most quickly in isolation [10,15,16]. Phillips & Perkins [10] have introduced the concept of 'spatio-temporal fitness' to explain why evolution during invasions selects for a combination of dispersal and reproductive fitness. They do not consider mutation and therefore predict that the product of dispersal and reproductive fitness (or, equivalently, monomorphic invasion speed) is maximized. Our results resemble those of Laroche *et al.* [18], who showed that evolution in metacommunities can lead to mixed dispersal strategies in the presence of trade-offs. However, the promotion of the densities of the vanguard strains is rather different from the evolution of a mixed strategy. First, while one vanguard strain leads the invasion, the density of the other one can trail behind non-vanguard strains (figures 1*b,d* and 2*b*) so the species is not dominated by these two strains. Second, the vanguard strains do not represent competing but rather cooperative strategies. We do not expect that anomalous invasion dynamics can lead to evolutionary branching between the vanguard strains since mutation between the vanguard strains is essential.

The anomalous dynamics we report require a combination of dispersal polymorphism, dispersal-fitness trade-off, and mutation during invasions. Without the trade-off, i.e. if all strains have the same population growth rate [20,21], the invasion accelerates to the monomorphic speed of the most dispersive strain. Mutation is necessary because otherwise the strain with the fastest monomorphic speed leaves the others behind [16,17]. However, our results contrast with other studies on the effect of mutation on invasion dynamics, because anomalous invasion dynamics do not converge to monomorphic dynamics when the mutation rate becomes very small. Griette *et al.* [33] computed the mutation-dependent invasion speed

in a related 2-strain model, but without dispersal polymorphism so their invasion speeds converge to the fastest single-strain speed when the mutation rate approaches zero.

Our results predict that invasion dynamics depend critically on the curvature of the trade-off curve between dispersal and population growth rate, but while the existence of such trade-offs has been established [34–36] their shape has not been quantified empirically in much detail. This trade-off arises from the organism diverting resources either to dispersal or to reproductive success, but the rates describing these abilities depend on the details of the organism's physiology so could in principle take many different shapes. One plausible assumption would be that the organism's reproductive success is proportional to the energy diverted to reproductive organs, and the distance of each dispersal event is proportional to the energy diverted to movement organs, which would imply a straight-line trade-off between fitness and dispersal distance. However, while population growth rate is directly proportional to fitness, the diffusion constant is proportional to the square of the dispersal distance, which would imply that the $D(r)$ curve would be a parabola with positive curvature. On the other hand, diminishing returns would imply that an incremental improvement in one trait comes at a much greater cost when that trait is at the higher end of its range of possible values, which would suggest a trade-off with negative curvature (e.g. figure 3e) or possibly a more complex curve such as in figure 3f. Anomalous speeds could also occur when the trade-off curve has *negative* curvature, provided the species exists in distinct morphs (such as wing dimorphic crickets [37]) so that the trade-off is not a single continuous curve (and that the virtual strain with the fastest invasion speed lies between different segments of the trade-off curve). Thus, while anomalous speeds can be expected in a wide range of scenarios, to predict the species in which they occur we would need more detailed measurements of the trade-offs between dispersal and reproductive fitness than are currently available.

Our models may be simple and generic, but we expect the predictions to apply in a wide range of scenarios. Our model assumes a haploid species, but we predict our theory to hold in any species where offspring inherit their strain (with rare mutation) from only one parent when population densities are low. This is true for microbes as well as self-fertile plants. We used Brownian motion to model dispersal (i.e. assuming dispersal comprises many very short steps), but the analysis and our predictions would be similar if dispersal followed a jump process with thin-tailed (exponentially bounded) dispersal kernels. Our theory should apply to modestly fat-tailed dispersal kernels of the class which lead to finite speed waves (i.e. those which are not exponentially bounded but have finite second moment), but, at first sight, not to 'fatter' tailed kernels used to describe dispersal combining local movement and long-distance events [38–40]. Such kernels have been shown to predict accelerating invasions rather than a travelling wave of constant speed. However, it has been shown that fat-tailed dispersal kernels can describe a population of individuals, each performing Brownian motion but with a distribution of diffusion constants [41]. This is analogous to the dispersal polymorphism we discuss, but for the case of no trade-off between $r$ and $D$. In that case, and (unrealistically, but in common with [41]) assuming no upper limit to $D$, we also would predict invasions that accelerate without limit. However, if the more dispersive strains have a lower population growth

rate, then our theory predicts that the invasion could follow an asymptotic constant invasion speed determined by the shape of this trade-off. This shows that it could be misleading to characterize a species by a single dispersal kernel, without considering whether intrinsic dispersal ability or reproductive ability might differ among individuals. We expect a more complex theory is needed to account for other factors such as nonlinear diffusion or landscape heterogeneity, but our results constitute the first steps in this direction.

To observe anomalous invasion speeds empirically, a study species would need to have at least two strains which differ in their dispersal, where the stronger disperser has weaker population growth. It needs to be possible for these strains to mutate into each other (e.g. they represent single nucleotide polymorphisms). Also, mutation rates and/or population sizes need to be high enough that both strains will be present in the leading edge of the wave. This suggests that microbes or viruses are the most likely systems to display anomalous speeds, including pathogens where hosts constitute demes within a large local population. RNA viruses can have mutation rates as high as $10^{-3} - 10^{-5}$ per base pair per generation [42], and can exist at densities of $10^{12}$ individuals per gram of human faeces [43], so within-host populations could easily be high enough to generate anomalous invasion speeds. To maximize the 'speedup' (the ratio of the dimorphic speed to the faster monomorphic speed) the strains should differ as much as possible in their traits ($r_1$, $D_1$) and ($r_2$, $D_2$). If one strain were effectively immobile ($D_1 \to 0$), then equation (3.3) requires $r_1 > 2r_2$, and from equations (3.1) and (3.2) and the speedup is $\rho/(2\sqrt{\rho - 1})$, where $\rho = r_1/r_2$. This speedup can be large if $r_2$ is very small, but $r_2$ should not be so small that none of its offspring mutate into the other strain in the leading edge of the wave.

Another much-studied aspect of evolution during invasions is 'expansion load' [44], a progressive decrease in fitness and/or the speed of invasion or range expansion [4] due to the fixation of deleterious mutations near the invasion front. Evolution of dispersal can act in combination and remove the deceleration caused by expansion load [45]. However, it is not clear that anomalous invasion speeds can occur in populations where expansion load is significant. This is because the former requires large population sizes in the leading edge of the front, but the latter required populations small enough for deleterious mutations to become fixed. These two phenomena therefore represent complementary potential outcomes for evolution during range expansions.

Many global threats are due to biological invasions: food security from crop pests, antibiotic resistance, local disease outbreaks, and loss of biodiversity. Many other important phenomena can be modelled with similar equations, e.g. the cell population dynamics of tumour growth. Our results show that it is not safe to model an invading polymorphic population with a single 'average' strain, or as multiple strains that vary in a single trait. Nevertheless, we show that the invasion speed can be straightforwardly predicted from the dispersal-fitness trade-off alone. Our results highlight the importance of understanding and quantifying such trade-offs, as well as the need to account for evolution when considering central questions in ecology.

Data accessibility. Simulation software, simulation data, and code used to generate figures are available as part of the electronic supplementary material.

Authors' contributions. Both authors designed and performed the study, and both authors wrote the manuscript.

**Competing interests.** We declare we have no competing interests.
**Funding.** V.A.K. was supported by a PhD studentship from the Natural Environment Research Council (NERC ACCE DTP, grant no. NE/L002450/1).

**Acknowledgements.** We thank Mike Begon, Andy Fenton, Ilik Saccheri, and Ben Phillips for helpful comments on this manuscript. We also thank the Plant Health Centre Scotland for allowing V.A.K. time to complete edits of this manuscript outside of his usual role.

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
