## [Reviewer comments · Proceedings of the Royal Society B: Biological Sciences]

Review History

RSPB-2020-0451.R0 (Original submission)

Review form: Reviewer 1

Recommendation

Accept with minor revision (please list in comments)

Scientific importance: Is the manuscript an original and important contribution to its field?

Excellent

General interest: Is the paper of sufficient general interest?

Good

Quality of the paper: Is the overall quality of the paper suitable?

Excellent

Is the length of the paper justified?

Yes

Should the paper be seen by a specialist statistical reviewer?

No

Do you have any concerns about statistical analyses in this paper? If so, please specify them explicitly in your report.

No

It is a condition of publication that authors make their supporting data, code and materials available - either as supplementary material or hosted in an external repository. Please rate, if applicable, the supporting data on the following criteria.

Is it accessible?

N/A

Is it clear?

N/A

Is it adequate?

N/A

Do you have any ethical concerns with this paper?

No

Comments to the Author

This is an extremely interesting paper that makes a significant contribution to the theory of biological invasions, a subject with great current importance. The authors make use of new mathematical results to derive some very surprising results on the way in which genetic mutations can impact on invasion speeds. I went carefully through both the paper and the electronic supplement, and I am happy with both the mathematical calculations and the biological implications that the authors draw from them. In my opinion the importance of the results is such that they merit publication in a high profile journal such as Proceedings B. I have a few suggestions for minor changes (below), but subject to these I recommend that the paper be accepted for publication.

Minor changes:

1. P6: this is a very picky mathematical point, but I suggest removing the brackets around "spatially uniform". There might be spatially non-uniform equilibria as well -- though they are not relevant to the work in the paper.
2. P6 Paragraph beginning "We consider.....". The authors are slightly over-stating the results of Girardin (2018). As far as I am aware, that paper does not prove convergence to travelling waves. Rather it proves that c^* is the long-term spreading speed. This is not the same thing. For example a travelling front moving with the oscillating speed $(1+\sin(Kt))c^*$ has a spreading speed c^* (in the sense of Girardin) but is certainly not a travelling wave. (I am not suggesting that such oscillating fronts occur in the system being considered, of course). This is a mathematical nicety which has no implications for the way in which the Keenan and Cornell use Girardin's results, but some tightening of language is required.
3. P10, paragraph beginning "It is important.....". I think that the authors are omitting an important point in their discussion of the μ tends to zero limit. When μ is sufficiently small the model will break down because the densities of some strains in the front will be small enough that a continuum model is no longer appropriate.
4. P16 The authors write "This suggests that accelerating invasions are a sign that the species has not evolved to the biological limits of the trade-off.....". I disagree with this statement. There are a number of other unrelated potential explanations for an accelerating invasion, notably long-range dispersal with a fat-tailed kernel (mentioned elsewhere in the paper) and local dynamics with zero linear component (such as in the PDE $u_t = u_{xx} + u^2(1-u)$).

5. I felt that there was a notable omission from the discussion section, namely that there was no discussion of experimental testing. To me, this paper calls out for follow-up experimental work -- perhaps with a bacterial system. I would encourage the authors to lay out in as much detail as possible a plan for such experimental testing.

Review form: Reviewer 2

Recommendation

Major revision is needed (please make suggestions in comments)

Scientific importance: Is the manuscript an original and important contribution to its field?

Good

General interest: Is the paper of sufficient general interest?

Good

Quality of the paper: Is the overall quality of the paper suitable?

Marginal

Is the length of the paper justified?

No

Should the paper be seen by a specialist statistical reviewer?

No

Do you have any concerns about statistical analyses in this paper? If so, please specify them explicitly in your report.

No

It is a condition of publication that authors make their supporting data, code and materials available - either as supplementary material or hosted in an external repository. Please rate, if applicable, the supporting data on the following criteria.

Is it accessible?

Yes

Is it clear?

N/A

Is it adequate?

N/A

Do you have any ethical concerns with this paper?

No

Comments to the Author

Report on the manuscript entitled "Anomalous invasion dynamics due to dispersal polymorphism and dispersal-reproduction trade-offs" submitted by V. Keenan and S. Cornell to Proc B.

In this manuscript, the authors extend a previous model (Elliot and Cornell 2012) for the spatial spread of a population with two strains to an arbitrary (but finite) number of strains. They

present a reaction-diffusion model and a corresponding stochastic simulation model. They find an approximate formula for the spreading speed in the case of vanishingly small mutation rates. They find that the parameters of at most two strains determine the speed at which the population spreads. They find what they call an "anomalous spreading speed" (comments later). They also discuss how a potential dispersal-fecundity trade-off affects whether there can be an anomalous spreading speed. The results are illustrated with a few carefully chosen simulation results.

The manuscript contains a number of incorrect mathematical statements. I believe that they are fixable, but this requires work. I have some issues with the presentation of the results. I had to read the manuscript and the supplementary material several times before I understood what the actual results were. I find that the authors overstate some of their claims as well. All in all, the manuscript would benefit greatly from rewriting several parts and focusing the points better. The discussion is very broad; I believe that things can be shortened there as well. The authors also missed some relevant developments in the literature. I hope that my points below will give the authors ideas for how I think the manuscript could be strengthened. Overall, I think that the manuscript should be rejected because there is so much work to do. If the authors want to resubmit, I leave it to the editor to decide whether Proc B is the right venue or whether maybe a more specialized journal (e.g. Theoretical Ecology) would be more appropriate.

The following comments are in no particular order (perhaps slightly chronological).

1) What the authors call an "anomalous spreading speed" is NOT an anomalous spreading speed in the sense of Weinberger et al 2007, which they cite. I went back to the earlier Elliot and Cornell (2012) and realized that the authors had used the term there, also inconsistently with Weinberger et al. The definition in Weinberger is subtle. I quote from the abstract of their paper: "By this we mean that this species spreads at a speed which is strictly greater than its spreading speed in isolation from the other species and the speeds at which all the other species actually spread." The present authors, however, define it as "faster than any of the strains in isolation". In other words, the Weinberger definition requires and implies that when all compartments spread together, not all spread at the same rate. But in the way the present authors use the term, all compartments spread at the same rate when together.

A particular implication of Weinberger et al's definition is that one cannot determine the spreading speed from the linearization at zero. The present authors, however, base all their spreading speed calculations on linearization at zero. An actual proof of anomalous in the sense of Weinberger is much much more complicated than the calculations based on linearization that the authors present.

I am not saying that their calculations are incorrect. In fact, Weinberger et al in their paper point out that if the linearization at zero is irreducible, then the linearization gives the correct speed. This was proved by Lui in the 1990s. The introduction of Weinberger et al 2007 gives an excellent account of the results, the subtleties, and the differences between their work and Lui's. The model in the present manuscript has the property that the linearization is irreducible, and hence the calculation is correct. But the speed is not anomalous in the sense of Weinberger because all strains spread at the same rate, whereas the definition of "anomalous" by Weinberger requires that at least one strain spreads faster than the other strains when all spread together, not only in isolation.

It is unfortunate that this conceptual misunderstanding of the authors has been able to persist for 8 years since the 2012 paper. It makes me wonder how much or how little interaction there really is between the more mathematical and more ecological range of the spectrum of interdisciplinary research.

2) Line 37: either "can play" or "plays" but not "can plays"

3) Line 40: What does "significantly faster" mean? Order of magnitude? Are there estimates? With the parameters chosen by the authors, the difference is on the order of 20 per cent. Are the chosen parameters representative of any particular system? Could the authors present their results with parameters defensible for organisms that the authors claim are most likely to apply to this reaction-diffusion framework? And are the error bars in parameter estimates much smaller than the difference between the regular and the increased speed?

4) Lines 40-44: This is more of a stylistic recommendation. Instead of "However, it is not clear how to generalize...", I would suggest a different tone. Because it is pretty obvious how to generalize the model. Other people have done the mathematical analysis. The unclear things are other things. So I suggest some formulation along the lines: "When we generalize this situation to more than two strains, several new questions emerge. For example... "

5) Line 48: The authors write: "Here we develop a general theory for invasions by species with dispersal polymorphism". This is one of the places that I would consider a clear exaggeration that discredits all the research done before and in parallel. The authors don't develop the theory. They apply existing theory and extend existing models and find some new results. Yes. But they don't develop a new general theory. Maybe some journals encourage such overstatements, but personally, I feel turned off. The foundations of this theory go back to Lui in the 1990s and maybe even before, and there are many past and current aspects left out. A general theory should do much more than answer the question of how many strains determine the population spreading speed.

6) Line 64: the authors claim to derive exact results, but they are actually asymptotic results, as the authors point out in several places.

7) Line 76: the clause "after enough generations" is unnecessary and potentially misleading. There is a positive probability of mutation in any reproduction event/any generation.

8) Line 77/78: The authors claim that the nonspatial dynamics of their system (spatially constant solutions) all converge to a positive steady state as long as all mutation rates are positive. This statement is false. It is already false in the two-species case. This is easy to see: we consider a case of two species competition of Lotka-Volterra form (without mutation) and choose parameters such that the system is in the founder control regime, i.e., the semi-trivial states are locally stable and the coexistence state is unstable. Then we introduce mutation between the types at some constant rate μ . If $\mu=0$, we have the previous case, i.e. an unstable coexistence state. When μ is positive but small, the stability of the coexistence state does not change (by continuity). Instead, there are two locally stable states with both entries positive.

When there are more than two species, the long-term dynamics of a Lotka-Volterra system become even more complicated. There are periodic orbits and heteroclinic cycles and much more. And all of these persist under small mutation perturbations, the case that the authors study. The authors seem to not have read their math papers carefully. If they had looked closely into the Girardin 2018 paper, they would have seen that the situation is more subtle (just like with Weinberger et al 2007). I hope that the Elliot and Cornell 2012 paper does not contain the same mistake.

9) Line 80: delete "For this class of partial differential equations", because of the later referral to "equations of the form (1)". The statement of the sentence is incorrect. Girardin did not prove that these initial conditions develop into traveling waves, but that traveling wave solutions exist. This does not mean that they are stable.

10) The statement "This result can be obtained by linearizing..." is incorrect. The result by Girardini cannot be obtained by linearizing. It would not be worth a 60+ page paper if it could. What is true is that c^* can be obtained by linearization.

11) Line 84-86: The sentence: "However, before Girardin's proof..." should be changed. The statement should be "Girard showed that the spreading speed is linearly determined." That sentence should come *before* the preceding sentence, because the preceding sentence somehow explains what "linearly determined" means. The fact that the result was not known before Girardin is correct but why is it important to the reader and the authors? What would change if the result had been known already in 2000, say?

12) Line 87/88: "More details of the steps..." The authors have not told us anything about their calculations yet. Why is the pointer to the appendix here? This should go after the authors state their results. At present, we are still in the "methods" section.

13) I don't like the formulation of the model in (2). If μ is the mutation probability, then the reproduction term should be $(1+r_i * (1-\mu))$ and if mutations are equally likely to all possible strains (as the authors seem to assume here but not in model (1)), then the term in front of the summation sign should be μ/N . The way the authors write the equation, the term $(1-N * \mu)$ can become negative simply by adding more strains. Since the authors use μ in the continuous-time model as well (in the appendix) they should point out that there is it a rate whereas here it is a probability.

14) Line 90. The sentence should end after the word "dispersal". A new thought starts with the introduction of stochasticity.

15) Line 94/95: "We expect that ..." The models are different. Period. The authors can expect that some of the results that they see in the simulations are similar in the limit.

16) line 96: Results. There is a bit of a disconnect here. In the methods, the authors describe the model, in the results they start with a result, but nowhere do they actually describe what they did. It would be really nice to have at least one sentence to that effect in the main text so that the reader knows roughly what to expect from the appendix.

As for the presentation of the results, a different order would work better for me. First, the authors could state that all of their results are derived in the limit as μ approaches zero. Then they could state that the asymptotic spreading speed is the maximum of the expressions in (5) and (6), because this is the more general result. And then they could state the consequences of the result, namely that (i) an "anomalous" speed is possible, and (ii) that the population spread rate is determined by the parameter values of at most two strains.

There seem to be two relatively simple potential explanation as to why only two strains determine the invasion speed. One would be that there are only two parameters determining the speed: r and D . The other would be that if one takes the maximum of any finite number of functions, then, generically, at most two functions define what that maximum is. (By generically, I mean that an intersection typically occurs between two and almost never between three functions.) The authors could think about which of these (if any) applies and offer it to the reader as a more general insight.

17) The caption of Figure 1 contains reference to variables that are not listed in the main text. The caption should also indicate how the "speed" was measured. It is unclear why the authors use complicated numerical simulations of the PDE system to determine the speed when they have an exact formula, given by Girardin.

18) The authors discuss the frequency of the various strains at the edge of the invasion front. But they do it only in the one example without any general theory. It seems to me that the mathematics allows much stronger statements. The traveling wave profile to the linearized equation is determined by a certain eigenvector, and this eigenvector, suitably normalized, should give the relative frequencies of the strains at the front.

19) Line 135: "The analysis predicts that..." This sentence is correct, but it is not the authors' analysis that predicts that. This is the fundamental contribution of Girardin. The authors' work uses this result but does not prove it. The authors should be clear about whose result this is. The conclusion in the following sentence also falls into that category. It also relates to my comment 8).

20) Line 148-159: The authors need to completely re-write this paragraph. Maybe it is surprising that a small amount of mutation is sufficient to generate speeds higher than individual spreading speeds, but the result is not new. The result is essentially discussed in Weinberger et al 2007, for example. It is true that the particular examples that Weinberger et al give have no small parameter. But it is clear from their description that this phenomenon occurs as soon as the operator is irreducible. And it is clear that only a small perturbation is needed to make an operator irreducible. Weinberger et al even explain the mechanism: because of the diffusion operator, species' densities are positive everywhere. Since the front is pulled, its speed is determined at low density, hence, the fact that quantities are small does not matter. Furthermore, there is a recent example of the same phenomenon in Chapter 14.7 in the book on Integro-difference equations by Lutscher. Proposition 14.9 calculates the exact spreading speed (in discrete time) for a mix of two strains and shows that this is independent of the mutation rate, not just for small mutation rates.

21) Line 185: the clause starting with "when" does not make sense to me. In general, I would appreciate if the authors could make some of their sentences shorter.

22) Discussion:

Instead of "We have presented a general theory" a more appropriate statement would be "We have extended existing 2-strain theory to N strains"

The use of "anomalous" should be explained and distinguished from the original Weinberger meaning.

The invasion speed is insensitive to the mutation rates: only because the authors are considering the limit of vanishing mutations. They have shown no evidence that this result is true in general.

The fact that the speed does not depend on competition is correct, but it is not the result of these authors, it is a result by Girardin (and in some cases others earlier).

... even by strains that are outcompeted at equilibrium: Again, correct, but again not the result of the current authors. In fact, since the speed is linearly determined, the equilibrium (if it exists and is stable) does not enter the theory at all. This is not new at all. It has been known for some other models for a long time. The authors present none of the rich context of this topic but make it sound like it is their original result.

Line 223-225: I don't understand this statement. Makes no sense to me.

Line 225-227: This suggests that an accelerating... The author completely misses the wide literature on accelerating invasions due to certain dispersal characteristics.

Line 230: the surprise may be on the biological side, not on the mathematical side, as I outline above. On the other hand, it is known that results on spread rates in the pulled case depend sensitively on small things. So, maybe it should not be surprising at all. The authors could talk about the related result on a small fraction of individuals dispersing longer distances than the bulk (Reid's paradox). This is documented in many papers.

line 234: "than" should be "that". I don't think that this is so surprising, or at least it does not have to go unexplained, see comment 16). I think that most of the speculations Line 235 - 240 could be removed. The mentioning of density dependence should definitely be removed, see earlier

comment. The reference to the λ / k curve is not very helpful here. Why don't the authors follow conventional notation and call this function the dispersion relation already in the main text?

23) Later in the discussion, the authors speculate about deleterious mutations. Their model cannot encompass those since they would have $r_i < 0$.

24) References: Fisher and KPP should be capitalized in the reference Girardin.

SUPPLEMENTARY MATERIAL:

1) The sentence around equation (S.1) is way too long. I suggest a period at the end of the equation, then a new sentence.

2) There are a few steps in the model description that sound strange. Why, for example does it only work when the mutation rate is small so that μ_{ij}/b_i is less than one? The definition of the death rate includes r_i , but the definition of r_i then includes d_i again? Sounds circular.

3) While I agree that positive values for C_{ij} imply that these terms describe competitive interactions, whether the overall effect of mutation and competition is competitive or cooperative depends on population density.

4) Top of page 2: Replace "initially coexist" with "are initially present".

5) Second paragraph on page 2: This paragraph contains a number of errors. First: it is not true that a system of form (1) has a non-trivial stable equilibrium (see comments to main text). Second: it is not generally true that the solutions form traveling waves. It is only true that there exist traveling waves. It is also not true that the linearization was known only to be a lower bound. It is true that it is an upper bound. Because the system with nonlinearities is bounded by the system without nonlinearities, the corresponding speed of the nonlinear system cannot larger than for the linear system. This argument is true for Fisher's equation and can be found in a number of textbooks. The difficult part in the proofs is always to prove the lower bound. Weinberger 1982 did that for scalar systems, Lui's papers do that for cooperative systems. That's the first result in the series headed here, the Girardin is the most recent.

6) In (S.3) the min over k has to be and inf over $k > 0$. This is important.

7) The argument given at the top of page 3 is NOT sufficient to show that the limit with respect to μ can be exchanged with the min max. Here is an example. Define the function

$$f(k, \mu) = k^2 / (k^2 + \mu^2), k > 0.$$

Then taking the inf over k for fixed positive μ results in 0. Taking the limit in μ remains zero. On the other hand, taking the limit in μ first gives 1, independent of k , and taking the inf in k remains at 1.

There are other examples where spreading speeds do not depend continuously on a parameter. One example is the application to Reid's paradox. There are several references for this. A recent summary is in Chapter 12 in the book on integrodifference equations by Lutscher.

8) The main idea in Figure S1 is very close (I would argue essentially the same as) Figure 14.18 in the book on integrodifference equations by Lutscher.

10) Numerical simulations. While it is important to give the details on the numerical scheme, I don't understand why the author used numerical simulations at all when all the quantities can be obtained from exact equations. Numerical schemes always introduce errors.

Decision letter (RSPB-2020-0451.R0)

13-May-2020

Dear Dr Keenan,

I am writing to inform you that your manuscript RSPB-2020-0451 entitled "Anomalous invasion dynamics due to dispersal polymorphism and dispersal-reproduction trade-offs" has, in its current form, been rejected for publication in Proceedings B.

This action has been taken on the advice of referees, who have recommended that substantial revisions are necessary. With this in mind we would be happy to consider a resubmission, provided the comments of the referees are fully addressed. However please note that this is not a provisional acceptance.

Yours sincerely,
Professor Loeske Kruuk
<mailto:proceedingsb@royalsociety.org>

Associate Editor
Comments to Author:

I should apologise for my late assessment, I've been sitting on this for a bit too long... One of the reviewers is quite positive but the other is much more reserved. A part from presentational issues, the more critical reviewer raises the point that the present study is at odds with previous studies of the mechanism, and addresses a fundamental issue, namely whether linearisation at the front end of the invading wave is possible or not. This is not my daily study topic, so I cannot easily see where definitions issues play a role here or whether something more fundamental is going on. The more critical reviewer also asks if a revised MS would be sufficiently interesting for the Proceedings. There I could state that I wasn't aware of this mechanism at all, and whatever the details I think it should be interesting to a wide audience. To some, going from a two-strain analysis to an n-strain one may seem incremental, but in my view it really contributes something essential. So I think this MS should be considered. But this fundamental discrepancy with the

earlier studies must be elucidated before we can go any further and I agree with the reviewer that this will require more than cosmetic changes.

Reviewer(s)' Comments to Author:

Referee: 1

Comments to the Author(s)

This is an extremely interesting paper that makes a significant contribution to the theory of biological invasions, a subject with great current importance. The authors make use of new mathematical results to derive some very surprising results on the way in which genetic mutations can impact on invasion speeds. I went carefully through both the paper and the electronic supplement, and I am happy with both the mathematical calculations and the biological implications that the authors draw from them. In my opinion the importance of the results is such that they merit publication in a high profile journal such as *Proceedings B*. I have a few suggestions for minor changes (below), but subject to these I recommend that the paper be accepted for publication.

Minor changes:

1. P6: this is a very picky mathematical point, but I suggest removing the brackets around "spatially uniform". There might be spatially non-uniform equilibria as well -- though they are not relevant to the work in the paper.

2. P6 Paragraph beginning "We consider.....". The authors are slightly over-stating the results of Girardin (2018). As far as I am aware, that paper does not prove convergence to travelling waves. Rather it proves that c^* is the long-term spreading speed. This is not the same thing. For example a travelling front moving with the oscillating speed $(1+\sin(Kt))c^*$ has a spreading speed c^* (in the sense of Girardin) but is certainly not a travelling wave. (I am not suggesting that such oscillating fronts occur in the system being considered, of course). This is a mathematical nicety which has no implications for the way in which the Keenan and Cornell use Girardin's results, but some tightening of language is required.

3. P10, paragraph beginning "It is important....". I think that the authors are omitting an important point in their discussion of the μ tends to zero limit. When μ is sufficiently small the model will break down because the densities of some strains in the front will be small enough that a continuum model is no longer appropriate.

4. P16 The authors write "This suggests that accelerating invasions are a sign that the species has not evolved to the biological limits of the trade-off.....". I disagree with this statement. There are a number of other unrelated potential explanations for an accelerating invasion, notably long-range dispersal with a fat-tailed kernel (mentioned elsewhere in the paper) and local dynamics with zero linear component (such as in the PDE $u_t = u_{xx} + u^2(1-u)$).

5. I felt that there was a notable omission from the discussion section, namely that there was no discussion of experimental testing. To me, this paper calls out for follow-up experimental work -- perhaps with a bacterial system. I would encourage the authors to lay out in as much detail as possible a plan for such experimental testing.

Referee: 2

Comments to the Author(s)

Report on the manuscript entitled "Anomalous invasion dynamics due to dispersal polymorphism and dispersal-reproduction trade-offs" submitted by V. Keenan and S. Cornell to *Proc B*.

In this manuscript, the authors extend a previous model (Elliot and Cornell 2012) for the spatial spread of a population with two strains to an arbitrary (but finite) number of strains. They present a reaction-diffusion model and a corresponding stochastic simulation model. They find an approximate formula for the spreading speed in the case of vanishingly small mutation rates. They find that the parameters of at most two strains determine the speed at which the population spreads. They find what they call an "anomalous spreading speed" (comments later). They also discuss how a potential dispersal-fecundity trade-off affects whether there can be an anomalous spreading speed. The results are illustrated with a few carefully chosen simulation results.

The manuscript contains a number of incorrect mathematical statements. I believe that they are fixable, but this requires work. I have some issues with the presentation of the results. I had to read the manuscript and the supplementary material several times before I understood what the actual results were. I find that the authors overstate some of their claims as well. All in all, the manuscript would benefit greatly from rewriting several parts and focusing the points better. The discussion is very broad; I believe that things can be shortened there as well. The authors also missed some relevant developments in the literature. I hope that my points below will give the authors ideas for how I think the manuscript could be strengthened. Overall, I think that the manuscript should be rejected because there is so much work to do. If the authors want to resubmit, I leave it to the editor to decide whether Proc B is the right venue or whether maybe a more specialized journal (e.g. Theoretical Ecology) would be more appropriate.

The following comments are in no particular order (perhaps slightly chronological).

1) What the authors call an "anomalous spreading speed" is NOT an anomalous spreading speed in the sense of Weinberger et al 2007, which they cite. I went back to the earlier Elliot and Cornell (2012) and realized that the authors had used the term there, also inconsistently with Weinberger et al. The definition in Weinberger is subtle. I quote from the abstract of their paper: "By this we mean that this species spreads at a speed which is strictly greater than its spreading speed in isolation from the other species and the speeds at which all the other species actually spread." The present authors, however, define it as "faster than any of the strains in isolation". In other words, the Weinberger definition requires and implies that when all compartments spread together, not all spread at the same rate. But in the way the present authors use the term, all compartments spread at the same rate when together.

A particular implication of Weinberger et al's definition is that one cannot determine the spreading speed from the linearization at zero. The present authors, however, base all their spreading speed calculations on linearization at zero. An actual proof of anomalous in the sense of Weinberger is much much more complicated than the calculations based on linearization that the authors present.

I am not saying that their calculations are incorrect. In fact, Weinberger et al in their paper point out that if the linearization at zero is irreducible, then the linearization gives the correct speed. This was proved by Lui in the 1990s. The introduction of Weinberger et al 2007 gives an excellent account of the results, the subtleties, and the differences between their work and Lui's. The model in the present manuscript has the property that the linearization is irreducible, and hence the calculation is correct. But the speed is not anomalous in the sense of Weinberger because all strains spread at the same rate, whereas the definition of "anomalous" by Weinberger requires that at least one strain spreads faster than the other strains when all spread together, not only in isolation.

It is unfortunate that this conceptual misunderstanding of the authors has been able to persist for 8 years since the 2012 paper. It makes me wonder how much or how little interaction there really is between the more mathematical and more ecological range of the spectrum of interdisciplinary research.

2) Line 37: either "can play" or "plays" but not "can plays"

3) Line 40: What does "significantly faster" mean? Order of magnitude? Are there estimates? With the parameters chosen by the authors, the difference is on the order of 20 per cent. Are the chosen parameters representative of any particular system? Could the authors present their results with parameters defensible for organisms that the authors claim are most likely to apply to this reaction-diffusion framework? And are the error bars in parameter estimates much smaller than the difference between the regular and the increased speed?

4) Lines 40-44: This is more of a stylistic recommendation. Instead of "However, it is not clear how to generalize...", I would suggest a different tone. Because it is pretty obvious how to generalize the model. Other people have done the mathematical analysis. The unclear things are other things. So I suggest some formulation along the lines: "When we generalize this situation to more than two strains, several new questions emerge. For example..."

5) Line 48: The authors write: "Here we develop a general theory for invasions by species with dispersal polymorphism". This is one of the places that I would consider a clear exaggeration that discredits all the research done before and in parallel. The authors don't develop the theory. They apply existing theory and extend existing models and find some new results. Yes. But they don't develop a new general theory. Maybe some journals encourage such overstatements, but personally, I feel turned off. The foundations of this theory go back to Lui in the 1990s and maybe even before, and there are many past and current aspects left out. A general theory should do much more than answer the question of how many strains determine the population spreading speed.

6) Line 64: the authors claim to derive exact results, but they are actually asymptotic results, as the authors point out in several places.

7) Line 76: the clause "after enough generations" is unnecessary and potentially misleading. There is a positive probability of mutation in any reproduction event/any generation.

8) Line 77/78: The authors claim that the nonspatial dynamics of their system (spatially constant solutions) all converge to a positive steady state as long as all mutation rates are positive. This statement is false. It is already false in the two-species case. This is easy to see: we consider a case of two species competition of Lotka-Volterra form (without mutation) and choose parameters such that the system is in the founder control regime, i.e., the semi-trivial states are locally stable and the coexistence state is unstable. Then we introduce mutation between the types at some constant rate μ . If $\mu=0$, we have the previous case, i.e. an unstable coexistence state. When μ is positive but small, the stability of the coexistence state does not change (by continuity). Instead, there are two locally stable states with both entries positive.

When there are more than two species, the long-term dynamics of a Lotka-Volterra system become even more complicated. There are periodic orbits and heteroclinic cycles and much more. And all of these persist under small mutation perturbations, the case that the authors study.

The authors seem to not have read their math papers carefully. If they had looked closely into the Girardin 2018 paper, they would have seen that the situation is more subtle (just like with Weinberger et al 2007). I hope that the Elliot and Cornell 2012 paper does not contain the same mistake.

9) Line 80: delete "For this class of partial differential equations", because of the later referral to "equations of the form (1)". The statement of the sentence is incorrect. Girardin did not prove that these initial conditions develop into traveling waves, but that traveling wave solutions exist. This does not mean that they are stable.

10) The statement "This result can be obtained by linearizing..." is incorrect. The result by Girardini cannot be obtained by linearizing. It would not be worth a 60+ page paper if it could. What is true is that c^* can be obtained by linearization.

11) Line 84-86: The sentence: "However, before Girardin's proof..." should be changed. The statement should be "Girard showed that the spreading speed is linearly determined." That sentence should come *before* the preceding sentence, because the preceding sentence somehow explains what "linearly determined" means. The fact that the result was not known before Girardin is correct but why is it important to the reader and the authors? What would change if the result had been known already in 2000, say?

12) Line 87/88: "More details of the steps..." The authors have not told us anything about their calculations yet. Why is the pointer to the appendix here? This should go after the authors state their results. At present, we are still in the "methods" section.

13) I don't like the formulation of the model in (2). If μ is the mutation probability, then the reproduction term should be $(1+r_i * (1-\mu))$ and if mutations are equally likely to all possible strains (as the authors seem to assume here but not in model (1)), then the term in front of the summation sign should be μ/N . The way the authors write the equation, the term $(1-N * \mu)$ can become negative simply by adding more strains.

Since the authors use μ in the continuous-time model as well (in the appendix) they should point out that there is it a rate whereas here it is a probability.

14) Line 90. The sentence should end after the word "dispersal". A new thought starts with the introduction of stochasticity.

15) Line 94/95: "We expect that ..." The models are different. Period. The authors can expect that some of the results that they see in the simulations are similar in the limit.

16) line 96: Results. There is a bit of a disconnect here. In the methods, the authors describe the model, in the results they start with a result, but nowhere do they actually describe what they did. It would be really nice to have at least one sentence to that effect in the main text so that the reader knows roughly what to expect from the appendix.

As for the presentation of the results, a different order would work better for me. First, the authors could state that all of their results are derived in the limit as μ approaches zero. Then they could state that the asymptotic spreading speed is the maximum of the expressions in (5) and (6), because this is the more general result. And then they could state the consequences of the result, namely that (i) an "anomalous" speed is possible, and (ii) that the population spread rate is determined by the parameter values of at most two strains.

There seem to be two relatively simple potential explanation as to why only two strains determine the invasion speed. One would be that there are only two parameters determining the speed: r and D . The other would be that if one takes the maximum of any finite number of functions, then, generically, at most two functions define what that maximum is. (By generically, I mean that an intersection typically occurs between two and almost never between three functions.) The authors could think about which of these (if any) applies and offer it to the reader as a more general insight.

17) The caption of Figure 1 contains reference to variables that are not listed in the main text. The caption should also indicate how the "speed" was measured. It is unclear why the authors use complicated numerical simulations of the PDE system to determine the speed when they have an exact formula, given by Girardin.

18) The authors discuss the frequency of the various strains at the edge of the invasion front. But they do it only in the one example without any general theory. It seems to me that the mathematics allows much stronger statements. The traveling wave profile to the linearized equation is determined by a certain eigenvector, and this eigenvector, suitably normalized, should give the relative frequencies of the strains at the front.

19) Line 135: "The analysis predicts that..." This sentence is correct, but it is not the authors' analysis that predicts that. This is the fundamental contribution of Girardin. The authors' work *uses* this result but does not *prove* it. The authors should be clear about whose result this is. The conclusion in the following sentence also falls into that category. It also relates to my comment 8).

20) Line 148-159: The authors need to completely re-write this paragraph. Maybe it is surprising that a small amount of mutation is sufficient to generate speeds higher than individual spreading speeds, but the result is not new. The result is essentially discussed in Weinberger et al 2007, for example. It is true that the particular examples that Weinberger et al give have no small parameter. But it is clear from their description that this phenomenon occurs as soon as the operator is irreducible. And it is clear that only a small perturbation is needed to make an operator irreducible. Weinberger et al even explain the mechanism: because of the diffusion operator, species' densities are positive everywhere. Since the front is pulled, its speed is determined at low density, hence, the fact that quantities are small does not matter. Furthermore, there is a recent example of the same phenomenon in Chapter 14.7 in the book on Integro-difference equations by Lutscher. Proposition 14.9 calculates the exact spreading speed (in discrete time) for a mix of two strains and shows that this is independent of the mutation rate, not just for small mutation rates.

21) Line 185: the clause starting with "when" does not make sense to me. In general, I would appreciate if the authors could make some of their sentences shorter.

22) Discussion:

Instead of "We have presented a general theory" a more appropriate statement would be "We have extended existing 2-strain theory to N strains"

The use of "anomalous" should be explained and distinguished from the original Weinberger meaning.

The invasion speed is insensitive to the mutation rates: only because the authors are considering the limit of vanishing mutations. They have shown no evidence that this result is true in general.

The fact that the speed does not depend on competition is correct, but it is not the result of these authors, it is a result by Girardin (and in some cases others earlier).

... even by strains that are outcompeted at equilibrium: Again, correct, but again not the result of the current authors. In fact, since the speed is linearly determined, the equilibrium (if it exists and is stable) does not enter the theory at all. This is not new at all. It has been known for some other models for a long time. The authors present none of the rich context of this topic but make it sound like it is their original result.

Line 223-225: I don't understand this statement. Makes no sense to me.

Line 225-227: This suggests that an accelerating... The author completely misses the wide literature on accelerating invasions due to certain dispersal characteristics.

Line 230: the surprise may be on the biological side, not on the mathematical side, as I outline above. On the other hand, it is known that results on spread rates in the pulled case depend sensitively on small things. So, maybe it should not be surprising at all. The authors could talk

about the related result on a small fraction of individuals dispersing longer distances than the bulk (Reid's paradox). This is documented in many papers.

line 234: "than" should be "that". I don't think that this is so surprising, or at least it does not have to go unexplained, see comment 16). I think that most of the speculations Line 235 - 240 could be removed. The mentioning of density dependence should definitely be removed, see earlier comment. The reference to the λ / k curve is not very helpful here. Why don't the authors follow conventional notation and call this function the dispersion relation already in the main text?

23) Later in the discussion, the authors speculate about deleterious mutations. Their model cannot encompass those since they would have $r_i < 0$.

24) References: Fisher and KPP should be capitalized in the reference Girardin.

SUPPLEMENTARY MATERIAL:

1) The sentence around equation (S.1) is way too long. I suggest a period at the end of the equation, then a new sentence.

2) There are a few steps in the model description that sound strange. Why, for example does it only work when the mutation rate is small so that μ_{ij}/b_i is less than one? The definition of the death rate includes r_i , but the definition of r_i then includes d_i again? Sounds circular.

3) While I agree that positive values for C_{ij} imply that these terms describe competitive interactions, whether the overall effect of mutation and competition is competitive or cooperative depends on population density.

4) Top of page 2: Replace "initially coexist" with "are initially present".

5) Second paragraph on page 2: This paragraph contains a number of errors. First: it is not true that a system of form (1) has a non-trivial stable equilibrium (see comments to main text). Second: it is not generally true that the solutions form traveling waves. It is only true that there exist traveling waves. It is also not true that the linearization was known only to be a lower bound. It is true that it is an upper bound. Because the system with nonlinearities is bounded by the system without nonlinearities, the corresponding speed of the nonlinear system cannot larger than for the linear system. This argument is true for Fisher's equation and can be found in a number of textbooks. The difficult part in the proofs is always to prove the lower bound. Weinberger 1982 did that for scalar systems, Lui's papers do that for cooperative systems. That's the first result in the series headed here, the Girardin is the most recent.

6) In (S.3) the min over k has to be and inf over $k > 0$. This is important.

7) The argument given at the top of page 3 is NOT sufficient to show that the limit with respect to μ can be exchanged with the min max. Here is an example. Define the function

$$f(k, \mu) = k^2 / (k^2 + \mu^2), k > 0.$$

Then taking the inf over k for fixed positive μ results in 0. Taking the limit in μ remains zero. On the other hand, taking the limit in μ first gives 1, independent of k , and taking the inf in k remains at 1.

There are other examples where spreading speeds do not depend continuously on a parameter. One example is the application to Reid's paradox. There are several references for this. A recent summary is in Chapter 12 in the book on integrodifference equations by Lutscher.

8) The main idea in Figure S1 is very close (I would argue essentially the same as) Figure 14.18 in the book on integrodifference equations by Lutscher.

10) Numerical simulations. While it is important to give the details on the numerical scheme, I don't understand why the author used numerical simulations at all when all the quantities can be obtained from exact equations. Numerical schemes always introduce errors.

Author's Response to Decision Letter for (RSPB-2020-0451.R0)

See Appendix A.

RSPB-2020-2825.R0

Review form: Reviewer 2

Recommendation

Accept as is

Scientific importance: Is the manuscript an original and important contribution to its field?

Good

General interest: Is the paper of sufficient general interest?

Good

Quality of the paper: Is the overall quality of the paper suitable?

Good

Is the length of the paper justified?

Yes

Should the paper be seen by a specialist statistical reviewer?

No

Do you have any concerns about statistical analyses in this paper? If so, please specify them explicitly in your report.

No

It is a condition of publication that authors make their supporting data, code and materials available - either as supplementary material or hosted in an external repository. Please rate, if applicable, the supporting data on the following criteria.

Is it accessible?

N/A

Is it clear?

N/A

Is it adequate?

N/A

Do you have any ethical concerns with this paper?

No

Comments to the Author

I have no further comments

Decision letter (RSPB-2020-2825.R0)

02-Dec-2020

Dear Dr Keenan

I am pleased to inform you that your Review manuscript RSPB-2020-2825 entitled "Anomalous invasion dynamics due to dispersal polymorphism and dispersal-reproduction trade-offs" has been accepted for publication in Proceedings B.

The Associate Editor has recommended acceptance without any further changes. Therefore, please proof-read your manuscript carefully and upload your final files for publication. Because the schedule for publication is very tight, it is a condition of publication that you submit the revised version of your manuscript within 7 days. If you do not think you will be able to meet this date please let me know immediately.

To upload your manuscript, log into <http://mc.manuscriptcentral.com/prsb> and enter your Author Centre, where you will find your manuscript title listed under "Manuscripts with Decisions." Under "Actions," click on "Create a Revision." Your manuscript number has been appended to denote a revision.

You will be unable to make your revisions on the originally submitted version of the manuscript. Instead, upload a new version through your Author Centre.

- 1) A text file of the manuscript (doc, txt, rtf or tex), including the references, tables (including captions) and figure captions. Please remove any tracked changes from the text before submission. PDF files are not an accepted format for the "Main Document".
- 2) A separate electronic file of each figure (tiff, EPS or print-quality PDF preferred). The format should be produced directly from original creation package, or original software format. Please note that PowerPoint files are not accepted.
- 3) Electronic supplementary material: this should be contained in a separate file from the main text and the file name should contain the author's name and journal name, e.g `authorname_procb_ESM_figures.pdf`

All supplementary materials accompanying an accepted article will be treated as in their final form. They will be published alongside the paper on the journal website and posted on the online figshare repository. Files on figshare will be made available approximately one week before the accompanying article so that the supplementary material can be attributed a unique DOI. Please see: <https://royalsociety.org/journals/authors/author-guidelines/>

4) Data-Sharing and data citation

It is a condition of publication that data supporting your paper are made available. Data should be made available either in the electronic supplementary material or through an appropriate

repository. Details of how to access data should be included in your paper. Please see <https://royalsociety.org/journals/ethics-policies/data-sharing-mining/> for more details.

If you wish to submit your data to Dryad (<http://datadryad.org/>) and have not already done so you can submit your data via this link <http://datadryad.org/submit?journalID=RSPB&manu=RSPB-2020-2825> which will take you to your unique entry in the Dryad repository.

Once again, thank you for submitting your manuscript to Proceedings B and I look forward to receiving your final version. If you have any questions at all, please do not hesitate to get in touch.

Sincerely,
Professor Loeske Kruuk
<mailto:proceedingsb@royalsociety.org>

Reviewer(s)' Comments to Author:

Referee: 2
Comments to the Author(s).
I have no further comments

Sincerely,
Proceedings B
<mailto:proceedingsb@royalsociety.org>

Decision letter (RSPB-2020-2825.R1)

08-Dec-2020

Dear Dr Cornell

I am pleased to inform you that your manuscript entitled "Anomalous invasion dynamics due to dispersal polymorphism and dispersal-reproduction trade-offs" has been accepted for publication in Proceedings B.

Your article has been estimated as being 9 pages long. Our Production Office will be able to confirm the exact length at proof stage.

Open Access

Paper charges

Sincerely,

Appendix A

Author response to editor and reviewers comments RSPB-2020-0451 “Anomalous invasion dynamics due to dispersal polymorphism and dispersal-reproduction trade-offs”

In these responses, the comments of the editor and referees are given in *red italics*, and our responses given in black upright font.

Associate Editor

Comments to Author:

I should apologise for my late assessment, I've been sitting on this for a bit too long... One of the reviewers is quite positive but the other is much more reserved. A part from presentational issues, the more critical reviewer raises the point that the present study is at odds with previous studies of the mechanism, and addresses a fundamental issue, namely whether linearisation at the front end of the invading wave is possible or not. This is not my daily study topic, so I cannot easily see where definitions issues play a role here or whether something more fundamental is going on. The more critical reviewer also asks if a revised MS would be sufficiently interesting for the Proceedings. There I could state that I wasn't aware of this mechanism at all, and whatever the details I think it should be interesting to a wide audience. To some, going from a two-strain analysis to an n-strain one may seem incremental, but in my view it really contributes something essential. So I think this MS should be considered. But this fundamental discrepancy with the earlier studies must be elucidated before we can go any further and I agree with the reviewer that this will require more than cosmetic changes.

We thank the Associate Editor for their comments, and would like to state upfront that Referee 2 does not actually disagree with our results, or with the approach we use to calculate spreading speeds in our model (they say “*I am not saying that their calculations are incorrect*”). Their principal criticism is that we use the word “anomalous” in a slightly different way from a particular body of mathematical literature.

Where Referee 2 states that a linearization approach cannot be used to study “anomalous” invasion dynamics, they are referring to a more restrictive definition of this term (we will expand on this point later in these Responses, where we will argue that this more restrictive class of anomalous dynamics is unlikely to be ecologically relevant). The referee’s comments confirm that Girardin’s (2018) proof applies to the equations we are studying, i.e. that the correct spreading speed for our model is given by linearization.

“Anomalous” is just an adjective, and in the paper we define precisely the way we are using it. We believe it is the best word to describe the phenomenon we are reporting, and it is used in the same way elsewhere in the literature (e.g. M. Holzer, *Physica D* 270:1-10). At no point have we claimed to be commenting on the particular type of “anomalous” dynamics referred to by Referee 2. We do not believe that we need to be bound by the terminology used in a different discipline, any more than an article in a mathematical journal should be banned from using the word “species” in a less restrictive way to biologists.

We believe that the readership of Proceedings B will, in common with the Associate Editor, be surprised at our results and welcome the opportunity to resubmit our paper. We are grateful to both referees for their comments, which have allowed us to correct a number of details and improve the manuscript.

Reviewer(s)' Comments to Author:

Referee: 1

Comments to the Author(s)

This is an extremely interesting paper that makes a significant contribution to the theory of biological invasions, a subject with great current importance. The authors make use of new mathematical results to derive some very surprising results on the way in which genetic mutations can impact on invasion speeds. I went carefully through both the paper and the electronic supplement, and I am happy with both the mathematical calculations and the biological implications that the authors draw from them. In my opinion the importance of the results is such that they merit publication in a high profile journal such as Proceedings B. I have a few suggestions for minor changes (below), but subject to these I recommend that the paper be accepted for publication.

Minor changes:

1. P6: this is a very picky mathematical point, but I suggest removing the brackets around "spatially uniform". There might be spatially non-uniform equilibria as well -- though they are not relevant to the work in the paper.

We are happy to make this change. We now clarify that we are considering in the paper the scenario where there is a single stable equilibrium, whereas it is possible to construct parameter sets for our model where this is not the case. See also Referee 2's point 8.

2. P6 Paragraph beginning "We consider.....". The authors are slightly over-stating the results of Girardin (2018). As far as I am aware, that paper does not prove convergence to travelling waves. Rather it proves that c^ is the long-term spreading speed. This is not the same thing. For example a travelling front moving with the oscillating speed $(1+\sin(Kt))c^*$ has a spreading speed c^* (in the sense of Girardin) but is certainly not a travelling wave. (I am not suggesting that such oscillating fronts occur in the system being considered, of course). This is a mathematical nicety which has no implications for the way in which the Keenan and Cornell use Girardin's results, but some tightening of language is required.*

We thank the referee for pointing this out. We have re-worded the MS to state that Girardin's proof applies to the spreading speed, not to the fact that solutions are travelling waves (in the strict mathematical sense).

3. P10, paragraph beginning "It is important....". I think that the authors are omitting an important point in their discussion of the μ tends to zero limit. When μ is sufficiently small the model will break down because the densities of some strains in the front will be small enough that a continuum model is no longer appropriate.

The referee's point is essentially the same one that we make at the end of the results section, i.e. that demographic stochasticity means that the continuum model cannot be applied for arbitrarily small mutation rate (strictly, the referee refers to the granularity of individuals rather than stochasticity, but any reasonable model where individuals are granular will also be stochastic). We have added a note in this paragraph pointing the reader to the later material.

4. P16 The authors write "This suggests that accelerating invasions are a sign that the species has not evolved to the biological limits of the trade-off.....". I disagree with this statement. There are a number of other unrelated potential explanations for an accelerating invasion, notably long-range dispersal with a fat-tailed kernel (mentioned elsewhere in the

paper) and local dynamics with zero linear component (such as in the PDE $u_t = u_{xx} + u^2(1-u)$).

We have removed this sentence, deferring a discussion of accelerating fronts and fat-tailed kernels to the discussion.

5. I felt that there was a notable omission from the discussion section, namely that there was no discussion of experimental testing. To me, this paper calls out for follow-up experimental work -- perhaps with a bacterial system. I would encourage the authors to lay out in as much detail as possible a plan for such experimental testing.

Our original MS did contain a discussion (original page 20) of the conditions where anomalous speeds might be seen in real systems. We have now re-written this paragraph to make a clearer link to possible experimental testing, and discuss the characteristics that a study species would need to display measurable anomalous dynamics.

Referee: 2

Comments to the Author(s)

Report on the manuscript entitled "Anomalous invasion dynamics due to dispersal polymorphism and dispersal-reproduction trade-offs" submitted by V. Keenan and S. Cornell to Proc B.

In this manuscript, the authors extend a previous model (Elliot and Cornell 2012) for the spatial spread of a population with two strains to an arbitrary (but finite) number of strains. They present a reaction-diffusion model and a corresponding stochastic simulation model. They find an approximate formula for the spreading speed in the case of vanishingly small mutation rates. They find that the parameters of at most two strains determine the speed at which the population spreads. They find what they call an "anomalous spreading speed" (comments later). They also discuss how a potential dispersal-fecundity trade-off affects whether there can be an anomalous spreading speed. The results are illustrated with a few carefully chosen simulation results.

The manuscript contains a number of incorrect mathematical statements. I believe that they are fixable, but this requires work.

We are grateful for the referee for pointing out some mathematical details which need correcting. However, we note that none of these affect the material arguments or results of our paper, or the biological implications of the results.

I have some issues with the presentation of the results. I had to read the manuscript and the supplementary material several times before I understood what the actual results were.

It is common practice in modelling papers for a biological journal to defer the details of the calculation to supplementary material, leaving the paper to focus on the underlying model and the results. We have adopted this practice as we believe it makes our results most accessible to our readership.

I find that the authors overstate some of their claims as well.

This was not our intention, and we are happy to clarify this in the revision.

All in all, the manuscript would benefit greatly from rewriting several parts and focusing the points better. The discussion is very broad; I believe that things can be shortened there as well. The authors also missed some relevant developments in the literature.

The “relevant developments” referred to by the referee concern the mathematical literature on anomalous dynamics in a more restrictive sense than we are using it, in a different class of models from that which we consider. Since our focus is on the behaviour of polymorphic species described by a particular class of PDEs, rather than on the mathematics of anomalous invasions more broadly, we disagree that these developments are relevant to our paper.

I hope that my points below will give the authors ideas for how I think the manuscript could be strengthened.

While we are grateful to the referee for their suggestions, we have only adopted them where we believe they are appropriate for Proceedings B, as opposed to a more mathematical journal.

Overall, I think that the manuscript should be rejected because there is so much work to do. If the authors want to resubmit, I leave it to the editor to decide whether Proc B is the right venue or whether maybe a more specialized journal (e.g. Theoretical Ecology) would be more appropriate.

The following comments are in no particular order (perhaps slightly chronological).

1) What the authors call an "anomalous spreading speed" is NOT an anomalous spreading speed in the sense of Weinberger et al 2007, which they cite. I went back to the earlier Elliot and Cornell (2012) and realized that the authors had used the term there, also inconsistently with Weinberger et al. The definition in Weinberger is subtle. I quote from the abstract of their paper: "By this we mean that this species spreads at a speed which is strictly greater than its spreading speed in isolation from the other species and the speeds at which all the other species actually spread." The present authors, however, define it as "faster than any of the strains in isolation". In other words, the Weinberger definition requires and implies that when all compartments spread together, not all spread at the same rate. But in the way the present authors use the term, all compartments spread at the same rate when together.

We do not disagree, but we do not claim in our paper that we are using “anomalous” in the same sense as Weinberger. We state throughout that we use it to mean “faster than any strain in isolation”, and we argue that this is counterintuitive and interesting in its own right (to biologists, at least). We cite Weinberger et al at one point because invasions that are anomalous *sensu* Weinberger are also anomalous in our usage, but we do not claim that our anomalous invasion dynamics are anomalous *sensu* Weinberger.

To avoid any possible confusion, we have now added a note to the MS to clarify that Weinberger et al’s definition is more restrictive than ours. However, we reiterate that anomalous invasions *sensu* Weinberger et al is not the focus of our paper. Anomalous dynamics in the strict sense of Weinberger could not occur in any species where there is mutation between strains, because mutation would cause all strains to spread at the same speed.

A particular implication of Weinberger et al's definition is that one cannot determine the spreading speed from the linearization at zero. The present authors, however, base all their spreading speed calculations on linearization at zero. An actual proof of anomalous in the

sense of Weinberger is much much more complicated than the calculations based on linearization that the authors present.

We don't disagree, but this is not relevant to our paper because we are not discussing that particular type of anomalous dynamics. We freely admit that the mathematics we present is much simpler than that behind anomalous dynamics *sensu* Weinberger et al. However, we believe that the significance of a biological paper is determined by the implications of its results, rather than the methods used to obtain them.

I am not saying that their calculations are incorrect. In fact, Weinberger et al in their paper point out that if the linearization at zero is irreducible, then the linearization gives the correct speed. This was proved by Lui in the 1990s.

Our understanding was that Lui proved this result for purely cooperative equations, which does not include the class we consider. That is why Girardin's (2018) results are needed to prove that the spreading speeds we calculate are exact (as $t \rightarrow \infty$)

The introduction of Weinberger et al 2007 gives an excellent account of the results, the subtleties, and the differences between their work and Lui's. The model in the present manuscript has the property that the linearization is irreducible, and hence the calculation is correct. But the speed is not anomalous in the sense of Weinberger because all strains spread at the same rate, whereas the definition of "anomalous" by Weinberger requires that at least one strain spreads faster than the other strains when all spread together, not only in isolation.

Again, we do not disagree but this is not relevant to our paper since we are not claiming to add to the body of literature referred to by the referee. Moreover, we would argue that the type of anomalous dynamics *sensu* Weinberger— while undoubtedly very interesting mathematically — is very unlikely to be relevant to any biological system, where the fields represent populations of individuals. Anomalous dynamics *sensu* Weinberger is caused by the action of the slower species, even though, in the moving reference frame of the fastest species, these densities tend to zero as time increases. In real populations that are actually made up of individuals, beyond a certain point in time these densities will be actually zero and there is no process to accelerate the fastest species.

It is unfortunate that this conceptual misunderstanding of the authors has been able to persist for 8 years since the 2012 paper. It makes me wonder how much or how little interaction there really is between the more mathematical and more ecological range of the spectrum of interdisciplinary research.

2) Line 37: either "can play" or "plays" but not "can plays"

This is now corrected.

3) Line 40: What does "significantly faster" mean? Order of magnitude? Are there estimates?

What we meant to convey was that the speedup is of order 1, i.e. not just something which approaches zero when the mutation rate is small. The speedup can be arbitrarily large (see next point). We already discuss this point later in the MS, so since space is limited we have deleted the word "significantly" rather than explaining it at this point in the MS.

With the parameters chosen by the authors, the difference is on the order of 20 per cent. Are the chosen parameters representative of any particular system? Could the authors present their results with parameters defensible for organisms that the authors claim are most likely to apply to this reaction-diffusion framework? And are the error bars in parameter estimates much smaller than the difference between the regular and the increased speed?

The question of how much speedup is possible was discussed in Elliott and Cornell 2012. The speedup can be arbitrarily large, which can be seen from eqns (5) and (6). If r_i and D_j are made vanishingly small then both monomorphic speeds approach zero while the dimorphic speed remains finite, so the speedup (ratio of dimorphic to the fastest monomorphic speed) can be made arbitrarily large.

As we are presenting a generic framework, potentially applicable to a huge range of taxa, we don't think we can present a meaningful discussion of parameter values or confidence intervals at this stage. We do, however, note in our new discussion of experimental validation that the speedup is greatest for two strains that are as dissimilar as possible in dispersal and reproductive ability.

4) Lines 40-44: This is more of a stylistic recommendation. Instead of "However, it is not clear how to generalize...", I would suggest a different tone. Because it is pretty obvious how to generalize the model. Other people have done the mathematical analysis. The unclear things are other things. So I suggest some formulation along the lines: "When we generalize this situation to more than two strains, several new questions emerge. For example..."

We have adopted a new wording along the lines proposed by the referee. We agree that it's clear how to generalise the model, and that what is not obvious is what the results of such a generalisation would be.

5) Line 48: The authors write: "Here we develop a general theory for invasions by species with dispersal polymorphism". This is one of the places that I would consider a clear exaggeration that discredits all the research done before and in parallel. The authors don't develop the theory. They apply existing theory and extend existing models and find some new results. Yes. But they don't develop a new general theory. Maybe some journals encourage such overstatements, but personally, I feel turned off. The foundations of this theory go back to Lui in the 1990s and maybe even before, and there are many past and current aspects left out. A general theory should do much more than answer the question of how many strains determine the population spreading speed.

We have tempered our wording to "model" rather than "general theory", though we think our use of the term "general theory" is justified. We're not claiming to have a theory of everything, but we have developed a framework that documents and classifies all the scenarios for invasion by a large class of polymorphic species among a wide range of taxa. Our theory makes use of powerful mathematical results derived by others, but our contribution – showing how to predict the invasion speed from the geometry of the tradeoff curve – is by no means a trivial corollary, even though the mathematics we present is relatively simple.

We note that the referee uses the term "general theory" later in their review (point 18), implying that these words could be used to describe a calculation of the densities of the strains in the leading edge of the wave for a general set of growth, diffusion, and mutation rates. Our central results certainly qualify as a "general theory" in this sense.

6) Line 64: the authors claim to derive exact results, but they are actually asymptotic results, as the authors point out in several places.

Our results are “exact” in the sense that we calculate a particular quantity exactly – the value of the spreading speed in the limit where mutation rate tends to zero. We see no reason to change this wording since the sense in which the results are exact is clear later on.

7) Line 76: the clause "after enough generations" is unnecessary and potentially misleading. There is a positive probability of mutation in any reproduction event/any generation.

We disagree with the referee, and argue that this sort of wording is necessary. There are situations where mutation between certain strains is not possible within a single generation. In our “nearest neighbour” model, for instance, strain 1 can mutate into strain 2 but not into 3, 4, or 5. Nevertheless, after enough generations, the descendants of any one strain can be of any of the other strains.

8) Line 77/78: The authors claim that the nonspatial dynamics of their system (spatially constant solutions) all converge to a positive steady state as long as all mutation rates are positive. This statement is false. It is already false in the two-species case. This is easy to see: we consider a case of two species competition of Lotka-Volterra form (without mutation) and choose parameters such that the system is in the founder control regime, i.e., the semi-trivial states are locally stable and the coexistence state is unstable. Then we introduce mutation between the types at some constant rate $\nu\mu$. If $\nu\mu=0$, we have the previous case, i.e. an unstable coexistence state. When $\nu\mu$ is positive but small, the stability of the coexistence state does not change (by continuity). Instead, there are two locally stable states with both entries positive.

When there are more than two species, the long-term dynamics of a Lotka-Volterra system become even more complicated. There are periodic orbits and heteroclinic cycles and much more. And all of these persist under small mutation perturbations, the case that the authors study.

The referee is correct here. We should have stated instead that we are considering specifically the case where there is a single stable equilibrium, as the most ecologically relevant example for a single species with multiple strains; we had not intended to study all of the possibilities afforded by Lotka Volterra equations. We have revised our MS accordingly. Note that, as noted by Referee 1, the proofs of Girardin show that our results for the spreading speed will apply even if there is no stable equilibrium or travelling wave.

The authors seem to not have read their math papers carefully. If they had looked closely into the Girardin 2018 paper, they would have seen that the situation is more subtle (just like with Weinberger et al 2007). I hope that the Elliott and Cornell 2012 paper does not contain the same mistake.

Elliott and Cornell (2012) only treat the two-strain case, so periodic orbits, heteroclinic cycles etc. are not possible.

9) Line 80: delete "For this class of partial differential equations", because of the later referral to "equations of the form (1)". The statement of the sentence is incorrect. Girardin did not prove that these initial conditions develop into traveling waves, but that traveling wave solutions exist. This does not mean that they are stable.

We admit we had missed this point (made also by Referee 1), that Girardin calculated the spreading speed without proving that the solutions approach travelling waves. However, the existence of travelling waves for models with a single stable equilibrium invading an unstable

state is arguably uncontroversial, even if there is yet no formal proof. We now state that the existence of travelling waves is shown by simulations.

10) The statement "This result can be obtained by linearizing..." is incorrect. The result by Girardin cannot be obtained by linearizing. It would not be worth a 60+ page paper if it could. What is true is that c^ can be obtained by linearization.*

What we meant by "result" was the expression for c^* , not the proof that this expression is indeed the spreading speed. We mentioned this because the approach will be familiar to many theoretical ecologists, though they may be unfamiliar with Girardin's proof that it gives the exact spreading speed. We have re-worded this, and moved the comment that this procedure pre-dates Girardin's proof to the Supplementary Material

11) Line 84-86: The sentence: "However, before Girardin's proof..." should be changed. The statement should be "Girard showed that the spreading speed is linearly determined." That sentence should come before the preceding sentence, because the preceding sentence somehow explains what "linearly determined" means. The fact that the result was not known before Girardin is correct but why is it important to the reader and the authors? What would change if the result had been known already in 2000, say?

This material has now been re-worded to make it clearer (see point 10 above). We have not used the words "linearly determined" because this terminology will not be familiar to much of the readership of Proceedings B.

12) Line 87/88: "More details of the steps..." The authors have not told us anything about their calculations yet. Why is the pointer to the appendix here? This should go after the authors state their results. At present, we are still in the "methods" section.

The convention in biology that a calculation is part of the "methods" of a paper, so that only the end results (the simulation output and the expressions in eqns. (5)-(7)) belong in the results section. The Methods section is therefore the appropriate place to point the reader to the full calculation.

*13) I don't like the formulation of the model in (2). If μ is the mutation probability, then the reproduction term should be $(1+r_i * (1-\mu))$ and if mutations are equally likely to all possible strains (as the authors seem to assume here but not in model (1)), then the term in front of the summation sign should be μ/N . The way the authors write the equation, the term $(1-N * \mu)$ can become negative simply by adding more strains.*

Since the authors use μ in the continuous-time model as well (in the appendix) they should point out that there is it a rate whereas here it is a probability.

We have changed the name of the mutation parameter to η for the stochastic model, to avoid confusion with the continuous time model. However, there is no need to re-define the mutation parameter in the way suggested as our model is unambiguous, though we have included a note that $\eta < 1/N$

14) Line 90. The sentence should end after the word "dispersal". A new thought starts with the introduction of stochasticity.

This is now fixed.

15) Line 94/95: "We expect that ..." The models are different. Period. The authors can expect that some of the results that they see in the simulations are similar in the limit.

We have added a word to clarify that what we mean is that simulated realisations of the models are similar

16) line 96: Results. There is a bit of a disconnect here. In the methods, the authors describe the model, in the results they start with a result, but nowhere do they actually describe what they did. It would be really nice to have at least one sentence to that effect in the main text so that the reader knows roughly what to expect from the appendix.

This is addressed in the new methods section, where we state specifically how c^* is calculated and that we compute an expression valid for $\mu \rightarrow 0$.

As for the presentation of the results, a different order would work better for me. First, the authors could state that all of their results are derived in the limit as μ approaches zero. Then they could state that the asymptotic spreading speed is the maximum of the expressions in (5) and (6), because this is the more general result. And then they could state the consequences of the result, namely that (i) an "anomalous" speed is possible, and (ii) that the population spread rate is determined by the parameter values of at most two strains.

We think this is a matter of taste. We provide a sentence at the start of the paragraph to summarise the paragraph's content (i.e. that there are anomalous speeds), before detailing the steps that lead to that conclusion. The referee is suggesting that we should defer that information until the steps leading up to it are laid out. While the referee's approach might be more common in a mathematical journal, we think our approach of providing more "signposting" is more appropriate for a biological journal such as Proceedings B.

There seem to be two relatively simple potential explanation as to why only two strains determine the invasion speed. One would be that there are only two parameters determining the speed: r and D . The other would be that if one takes the maximum of any finite number of functions, then, generically, at most two functions define what that maximum is. (By generically, I mean that an intersection typically occurs between two and almost never between three functions.) The authors could think about which of these (if any) applies and offer it to the reader as a more general insight.

The referee's second explanation is correct, as illustrated by fig. S1 – the minimum of a function which is itself the maximum of several functions is determined by at most two of these functions (barring special cases which almost never happen, as stated by the referee). Our original paper already discussed this point (lines 235 and onwards in the original MS).

17) The caption of Figure 1 contains reference to variables that are not listed in the main text.

This is now corrected (ν_{ij} now defined in the methods).

The caption should also indicate how the "speed" was measured.

We disagree - these details (which are explained in the supplementary methods) are unimportant for understanding our results, so we do not believe that they belong in the main MS where space is limited.

It is unclear why the authors use complicated numerical simulations of the PDE system to determine the speed when they have an exact formula, given by Girardin.

It is standard practice to support mathematical results with model simulations in biological journals. Simulation results are more tangible to many readers, and are evidence that there isn't a mistake in the calculation. Moreover, our exact expressions only apply in the limit of long times and for vanishingly small mutation rate, so simulations are needed to show that they give a good approximation for finite times and mutation rates. The exact results also do not give the density profiles (and, as pointed out by both referees, Girardin's results do not even prove that the solutions are travelling waves).

18) The authors discuss the frequency of the various strains at the edge of the invasion front. But they do it only in the one example without any general theory. It seems to me that the mathematics allows much stronger statements. The traveling wave profile to the linearized equation is determined by a certain eigenvector, and this eigenvector, suitably normalized, should give the relative frequencies of the strains at the front.

While a detailed calculation of the densities at the leading edge of the travelling wave is possible, we do not believe that it would merit the space it would require in this paper. The densities themselves are not of particular interest apart from the issue of "spatial sorting" – previous authors have suggested that the more dispersive strains lead invasions by polymorphic species. Our aim is to show that spatial sorting does not always occur, and a single, numerical example (such as the one we give) is sufficient to make this point.

19) Line 135: "The analysis predicts that..." This sentence is correct, but it is not the authors' analysis that predicts that. This is the fundamental contribution of Girardin. The authors' work _uses_ this result but does not _prove_ it. The authors should be clear about whose result this is. The conclusion in the following sentence also falls into that category. It also relates to my comment 8).

We have modified the text to make it clearer that this result is due to Girardin's proof.

20) Line 148-159: The authors need to completely re-write this paragraph. Maybe it is surprising that a small amount of mutation is sufficient to generate speeds higher than individual spreading speeds, but the result is not new. The result is essentially discussed in Weinberger et al 2007, for example. It is true that the particular examples that Weinberger et al give have no small parameter. But it is clear from their description that this phenomenon occurs as soon as the operator is irreducible. And it is clear that only a small perturbation is needed to make an operator irreducible. Weinberger et al even explain the mechanism: because of the diffusion operator, species' densities are positive everywhere. Since the front is pulled, its speed is determined at low density, hence, the fact that quantities are small does not matter. Furthermore, there is a recent example of the same phenomenon in Chapter 14.7 in the book on Integro-difference equations by Lutscher. Proposition 14.9 calculates the exact spreading speed (in discrete time) for a mix of two strains and shows that this is independent of the mutation rate, not just for small mutation rates.

We are not clear what the referee is arguing here. There are two distinct ideas:

i. That mutation generates speeds that are faster than any strain in isolation.

ii. That the speedup relative to the case without mutation does not diminish to zero when the mutation approaches zero.

The first sentence of the paragraph in question makes it clear that it is concerned with point ii. However, the referee's criticisms seem to relate to point i. We are prepared to believe the Referee that it is clear from Weinberger et al. that a small perturbation can give rise to anomalous speeds of the type i. However, in the examples given by Weinberger et al a cooperative strength of order 1 produces a speedup of order 1. It is not clear from Weinberger et al (who do not discuss the case of small perturbations) whether a weak perturbation would produce a speedup of order 1, or whether this speedup would diminish to zero as the perturbation diminished to zero.

Moreover, we are not claiming that our MS provides the first example of anomalous speeds of type ii, as we make it clear throughout that this is present already in the two-strain case of Elliott and Cornell (2012). The example cited by the referee from Lutscher (2019) is essentially the same phenomenon as in Elliott and Cornell (2012) – the book describes that model as “a discrete-time version of the invasion model of two types (or morphs) from Elliott and Cornell (2012)”

We have moved this material to the Discussion section (2nd paragraph) to avoid the need to discuss this material twice, and we have modified the language to make it clear that we are discussing specifically point ii.

21) Line 185: the clause starting with "when" does not make sense to me. In general, I would appreciate if the authors could make some of their sentences shorter.

We have split this sentence into two, and tried to reword it to be clearer, but we are unsure why the referee is confused. In Fig. 3b, there are no valid dimorphic speeds because no two strains lie either side of the virtual strain, represented by the green triangle. The invasion speed is therefore the monomorphic speed of the fastest constituent strain. In Fig. 3a, the virtual strain represented by the green triangle does give a valid dimorphic speed because pairs of strains lie either side of it. However, in fig. 3a this virtual strain lies on the tradeoff curve and is therefore has the same traits as one of the constituent strains. Therefore, in both cases the invasion speed is the fastest monomorphic speed among the constituent strains (though in Fig. 3a this also happens to equal the valid dimorphic speed of a continuous set of pairs of constituent strains).

22) Discussion:

Instead of "We have presented a general theory" a more appropriate statement would be "We have extended existing 2-strain theory to N strains"

We have reworded as suggested

The use of "anomalous" should be explained and distinguished from the original Weinberger meaning.

We now make explicit reference to the fact that Weinberger's definition is more restrictive than ours.

The invasion speed is insensitive to the mutation rates: only because the authors are considering the limit of vanishing mutations. They have shown no evidence that this result is true in general.

We have clarified that this result only applies when mutation rates are small (which is the biologically interesting case).

The fact that the speed does not depend on competition is correct, but it is not the result of these authors, it is a result by Girardin (and in some cases others earlier).

We have added a citation to Girardin.

... even by strains that are outcompeted at equilibrium: Again, correct, but again not the result of the current authors. In fact, since the speed is linearly determined, the equilibrium (if it exists and is stable) does not enter the theory at all. This is not new at all. It has been known for some other models for a long time. The authors present none of the rich context of this topic but make it sound like it is their original result.

We are not claiming that we proved this result – we have made it clear that the proof is due to Girardin, and all we are doing is applying his results to our model. However, it is important to note that this will be surprising to many ecologists who do expect competition to affect invasion speeds (see for example Burton et al 2010).

Line 223-225: I don't understand this statement. Makes no sense to me.

It is a relic of an earlier draft where we discussed the properties of the monomorphic speeds, but these have in any case been discussed by other authors (Osnas et al 2015) and space limitations have led us to concentrate on dimorphic speeds. We still discuss the existence of accelerating invasions with fat-tailed dispersal kernels.

Line 225 227: This suggest that an accelerating... The author completely miss the wide literature on accelerating invasions due to certain dispersal characteristics.

See our response to Referee 1's point 4.

Line 230: the surprise may be on the biological side, not on the mathematical side, as I outline above. On the other hand, it is known that results on spread rates in the pulled case depend sensitively on small things. So, maybe it should not be surprising at all. The authors could talk about the related result on a small fraction of individuals dispersing longer distances than the bulk (Reid's paradox). This is documented in many papers.

Referee 1 describes our results as “surprising”, so we feel that we are justified in retaining this word. We are not claiming that no phenomenon of this type has been reported before – after all, the same phenomenon is reported for the two-strain case in Elliott and Cornell (2012) – but we maintain that these results will be counterintuitive for many readers, as evidenced by Referee 1's report. We do cite the issue of fat-tailed dispersal later on, and reference Reid's paradox through our citation of Clark et al.

line 234: "than" should be "that". I don't think that this is so surprising, or at least it does not have to go unexplained, see comment 16).

Again we feel that the word “surprising” is justified because of the reaction of Referee 1. As mentioned in our response to comment 16, this paragraph does contain an explanation for the phenomenon.

I think that most of the speculations Line 235 - 240 could be removed.

We disagree – it is essential that a paper contains a discussion of its possible ramifications and future work, so it would be remiss of us not to speculate what sorts of models might display the same behaviour.

The mentioning of density dependence should definitely be removed, see earlier comment.

We disagree. While we make use of Girardin’s proof, the fact that the speed depends only upon two strains is our own result.

The reference to the λ / k curve is not very helpful here.

We now use a built-up fraction to show that we mean λ divided by k (the phase velocity), not λ as a function of k (the dispersion relation). λ and k are defined in the Methods of the revised MS, and the reader is directed to the supplementary information for more details.

Why don't the authors follow conventional notation and call this function the dispersion relation already in the main text?

We do not use the term “dispersion relation” in the main text, and believe much of the readership of Proceedings B would not be familiar with it.

23) Later in the discussion, the authors speculate about deleterious mutations. Their model cannot encompass those since they would have $r_i < 0$.

The referee is incorrect here. A deleterious mutation is one that reduces fitness (and therefore r_i), but does not have to lead to negative growth rate.

24) References: Fisher and KPP should be capitalized in the reference Girardin.

Fixed

SUPPLEMENTARY MATERIAL:

1) The sentence around equation (S.1) is way too long. I suggest a period at the end of the equation, then a new sentence.

We have adopted the referee’s suggestion

2) There are a few steps in the model description that sound strange. Why, for example does it only work when the mutation rate is small so that μ_{ij}/b_i is less than one?

This is because mutation takes place at birth only; if $\mu_{ij}/b_i > 1$ then there are more “mutant births” than there are births in total.

The definition of the death rate includes r_i , but the definition of r_i then includes d_i again? Sounds circular.

We have re-written this to make it clearer.

3) While I agree that positive values for C_{ij} imply that these terms describe competitive interactions, whether the overall effect of mutation and competition is competitive or cooperative depends on population density.

We are using the standard ecological terminology here, where “interactions” are between individuals and are represented by nonlinear terms; in that usage a mutation is not an interaction and is neither “cooperative” or “competitive”. By contrast, mathematical terminology uses “cooperative” to mean that the rate of change of one density depends positively on the density of another. So, for example, density independent birth is a “cooperative interaction” to a mathematician, but is not an interaction at all to an ecologist.

Since Proceedings B is a biological journal, we think the ecological convention is more appropriate, and its readership will recognise “competitive interactions” as a synonym for C_{ij} positive.

4) Top of page 2: Replace "initially coexist" with "are initially present".

Done

5) Second paragraph on page 2: This paragraph contains a number of errors. First: it is not true that a system of form (1) has a non-trivial stable equilibrium (see comments to main text). Second: it is not generally true that the solutions form traveling waves. It is only true that there exist traveling waves. It is also not true that the linearization was known only to be a lower bound. It is true that it is an upper bound. Because the system with nonlinearities is bounded by the system without nonlinearities, the corresponding speed of the nonlinear system cannot larger than for the linear system. This argument is true for Fisher's equation and can be found in a number of textbooks. The difficult part in the proofs is always to prove the lower bound. Weinberger 1982 did that for scalar systems, Lui's papers do that for cooperative systems. That's the first result in the series headed here, the Girardin is the most recent.

These have now been corrected. We have removed the statement that linearization provides a lower bound (though this can be found in van Saarloos (2003)) as this is not relevant to our paper.

6) In (S.3) the min over k has to be and inf over $k > 0$. This is important.

Corrected.

7) The argument given at the top of page 3 is NOT sufficient to show that the limit with respect to μ can be exchanged with the min max. Here is an example. Define the function

$$f(k, \mu) = k^2 / (k^2 + \mu^2), \quad k > 0.$$

Then taking the inf over k for fixed positive μ results in 0. Taking the limit in μ remains zero. On the other hand, taking the limit in μ first gives 1, independent of k , and taking the inf in k remains at 1.

We now provide a more detailed and careful mathematical argument to show why we can (effectively) swap the order of these limits.

There are other examples where spreading speeds do not depend continuously on a parameter. One example is the application to Reid's paradox. There are several references for this. A recent summary is in Chapter 12 in the book on integrodifference equations by Lutscher.

We agree that this is an interesting example whose behaviour is similarly counterintuitive, and we now mention it in the Discussion.

8) The main idea in Figure S1 is very close (I would argue essentially the same as) Figure 14.18 in the book on integrodifference equations by Lutscher.

Our figure deals with three strains, whereas Lutscher fig. 14.18 deals with only two. The main idea predates Lutscher (2019) – it can be found in Morris et al (2019), available as a preprint in 2016 (arXiv:1612.06768). We now provide references to both.

10) Numerical simulations. While it is important to give the details on the numerical scheme, I don't understand why the author used numerical simulations at all when all the quantities can be obtained from exact equations. Numerical schemes always introduce errors.

See our answer to point 17.